# DeepPoseKit, a software toolkit for fast and robust animal pose estimation using deep learning

Jacob M Graving[1,2,3]*, Daniel Chae[4], Hemal Naik[1,2,3,5], Liang Li[1,2,3], Benjamin Koger[1,2,3], Blair R Costelloe[1,2,3], Iain D Couzin[1,2,3]*

[1]Department of Collective Behaviour, Max Planck Institute of Animal Behavior, Konstanz, Germany; [2]Department of Biology, University of Konstanz, Konstanz, Germany; [3]Centre for the Advanced Study of Collective Behaviour, University of Konstanz, Konstanz, Germany; [4]Department of Computer Science, Princeton University, Princeton, United States; [5]Chair for Computer Aided Medical Procedures, Technische Universität München, Munich, Germany

**Abstract** Quantitative behavioral measurements are important for answering questions across scientific disciplines—from neuroscience to ecology. State-of-the-art deep-learning methods offer major advances in data quality and detail by allowing researchers to automatically estimate locations of an animal's body parts directly from images or videos. However, currently available animal pose estimation methods have limitations in speed and robustness. Here, we introduce a new easy-to-use software toolkit, *DeepPoseKit*, that addresses these problems using an efficient multi-scale deep-learning model, called *Stacked DenseNet*, and a fast GPU-based peak-detection algorithm for estimating keypoint locations with subpixel precision. These advances improve processing speed >2x with no loss in accuracy compared to currently available methods. We demonstrate the versatility of our methods with multiple challenging animal pose estimation tasks in laboratory and field settings—including groups of interacting individuals. Our work reduces barriers to using advanced tools for measuring behavior and has broad applicability across the behavioral sciences.

*For correspondence:
jgraving@gmail.com (JMG);
icouzin@ab.mpg.de (IDC)

## Introduction

Understanding the relationships between individual behavior, brain activity (reviewed by *Krakauer et al., 2017*), and collective and social behaviors (*Rosenthal et al., 2015*; *Strandburg-Peshkin et al., 2013*; *Jolles et al., 2017*; *Klibaite et al., 2017*; *Klibaite and Shaevitz, 2019*) is a central goal of the behavioral sciences—a field that spans disciplines from neuroscience to psychology, ecology, and genetics. Measuring and modelling behavior is key to understanding these multiple scales of complexity, and, with this goal in mind, researchers in the behavioral sciences have begun to integrate theory and methods from physics, computer science, and mathematics (*Anderson and Perona, 2014*; *Berman, 2018*; *Brown and de Bivort, 2018*). A cornerstone of this interdisciplinary revolution is the use of state-of-the-art computational tools, such as computer vision algorithms, to automatically measure locomotion and body posture (*Dell et al., 2014*). Such a rich description of animal movement then allows for modeling, from first principles, the full behavioral repertoire of animals (*Stephens et al., 2011*; *Berman et al., 2014b*; *Berman et al., 2016*; *Wiltschko et al., 2015*; *Johnson et al., 2016b*; *Todd et al., 2017*; *Klibaite et al., 2017*; *Markowitz et al., 2018*; *Klibaite and Shaevitz, 2019*; *Costa et al., 2019*). Tools for automatically measuring animal movement represent a vital first step toward developing unified theories of behavior across scales (*Berman, 2018*; *Brown and de Bivort, 2018*). Therefore, technical factors like

**eLife digest** Studying animal behavior can reveal how animals make decisions based on what they sense in their environment, but measuring behavior can be difficult and time-consuming. Computer programs that measure and analyze animal movement have made these studies faster and easier to complete. These tools have also made more advanced behavioral experiments possible, which have yielded new insights about how the brain organizes behavior.

Recently, scientists have started using new machine learning tools called deep neural networks to measure animal behavior. These tools learn to measure animal posture – the positions of an animal's body parts in space – directly from real data, such as images or videos, without being explicitly programmed with instructions to perform the task. This allows deep learning algorithms to automatically track the locations of specific animal body parts in videos faster and more accurately than previous techniques. This ability to learn from images also removes the need to attach physical markers to animals, which may alter their natural behavior.

Now, Graving et al. have created a new deep learning toolkit for measuring animal behavior that combines components from previous tools with the latest advances in computer science. Simple modifications to how the algorithms are trained can greatly improve their performance. For example, adding connections between layers, or 'neurons', in the deep neural network and training the algorithm to learn the full geometry of the body – by drawing lines between body parts – both enhance its accuracy. As a result of adding these changes, the new toolkit can measure an animal's pose from previously unseen images with high speed and accuracy, after being trained on just 100 examples. Graving et al. tested their model on videos of fruit flies, zebras and locusts, and found that, after training, it was able to accurately track the animals' movements. The new toolkit has an easy-to-use software interface and is freely available for other scientists to use and build on.

The new toolkit may help scientists in many fields including neuroscience and psychology, as well as other computer scientists. For example, companies like Google and Apple use similar algorithms to recognize gestures, so making those algorithms faster and more efficient may make them more suitable for mobile devices like smartphones or virtual-reality headsets. Other possible applications include diagnosing and tracking injuries, or movement-related diseases in humans and livestock.

---

scalability, robustness, and usability are issues of critical importance, especially as researchers across disciplines begin to increasingly rely on these methods.

Two of the latest contributions to the growing toolbox for quantitative behavioral analysis are from *Mathis et al. (2018)* and *Pereira et al. (2019)*, who make use of a popular type of machine learning model called *convolutional neural networks*, or *CNNs* (*LeCun et al., 2015*; Appendix 2), to automatically measure detailed representations of animal posture—structural *keypoints*, or *joints*, on the animal's body—directly from images and without markers. While these methods offer a major advance over conventional methods with regard to data quality and detail, they have disadvantages in terms of speed and robustness, which may limit their practical applications. To address these problems, we introduce a new software toolkit, called *DeepPoseKit*, with methods that are fast, robust, and easy-to-use. We run experiments using multiple datasets to compare our new methods with those from *Mathis et al. (2018)* and *Pereira et al. (2019)*, and we find that our approach offers considerable improvements. These results also demonstrate the flexibility of our toolkit for both laboratory and field situations and exemplify the wide applicability of our methods across a range of species and experimental conditions.

## Measuring animal movement with computer vision

Collecting high-quality behavioral data is a challenging task, and while direct observations are important for gathering qualitative data about a study system, a variety of automated methods for quantifying movement have become popular in recent years (*Dell et al., 2014*; *Anderson and Perona, 2014*; *Kays et al., 2015*). Methods like video monitoring and recording help to accelerate data collection and reduce the effects of human intervention, but the task of manually scoring videos is time consuming and suffers from the same limitations as direct observation, namely observer bias and mental fatigue. Additionally, due to limitations of human observers' ability to process information,

many studies that rely on manual scoring use relatively small datasets to estimate experimental effects, which can lead to increased rates of statistical errors. Studies that lack the statistical resolution to robustly test hypotheses (commonly called 'power' in frequentist statistics) also raise concerns about the use of animals for research, as statistical errors caused by sparse data can impact researchers' ability to accurately answer scientific questions. These limitations have led to the development of automated methods for quantifying behavior using advanced imaging technologies (*Dell et al., 2014*) as well as sophisticated tags and collars with GPS, accelerometry, and acoustic-recording capabilities (*Kays et al., 2015*). Tools for automatically measuring the behavior of individuals now play a central role in our ability to study the neurobiology and ecology of animals, and reliance on these technologies for studying animal behavior will only increase in the future.

The rapid development of computer vision hardware and software in recent years has allowed for the use of automated image-based methods for measuring behavior across many experimental contexts (*Dell et al., 2014*). Early methods for quantifying movement with these techniques required highly controlled laboratory conditions. However, because animals exhibit different behaviors depending on their surroundings (*Strandburg-Peshkin et al., 2017*; *Francisco et al., 2019*; *Akhund-Zade et al., 2019*), laboratory environments are often less than ideal for studying many natural behaviors. Most conventional computer vision methods are also limited in their ability to accurately track groups of individuals over time, but nearly all animals are social at some point in their life and exhibit specialized behaviors when in the presence of conspecifics (*Strandburg-Peshkin et al., 2013*; *Rosenthal et al., 2015*; *Jolles et al., 2017*; *Klibaite et al., 2017*; *Klibaite and Shaevitz, 2019*; *Francisco et al., 2019*; *Versace et al., 2019*). These methods also commonly track only the animal's center of mass, which reduces the behavioral output of an individual to a two-dimensional or three-dimensional particle-like trajectory. While trajectory data are useful for many experimental designs, the behavioral repertoire of an animal cannot be fully described by its aggregate locomotory output. For example, stationary behaviors, like grooming and antennae movements, or subtle differences in walking gaits cannot be reliably detected by simply tracking an animal's center of mass (*Berman et al., 2014b*; *Wiltschko et al., 2015*).

Together these factors have driven the development of software that can accurately track the positions of marked (*Crall et al., 2015*; *Graving, 2017*; *Wild et al., 2018*; *Boenisch et al., 2018*) or unmarked (*Pérez-Escudero et al., 2014*; *Romero-Ferrero et al., 2019*) individuals as well as methods that can quantify detailed descriptions of an animal's posture over time (*Stephens et al., 2011*; *Berman et al., 2014b*; *Wiltschko et al., 2015*; *Mathis et al., 2018*; *Pereira et al., 2019*). Recently, these advancements have been further improved through the use of deep learning, a class of machine learning algorithms that learn complex statistical relationships from data (*LeCun et al., 2015*). Deep learning has opened the door to accurately tracking large groups of marked (*Wild et al., 2018*; *Boenisch et al., 2018*) or unmarked (*Romero-Ferrero et al., 2019*) individuals and has made it possible to measure the body posture of animals in nearly any context—including 'in the wild' (*Nath et al., 2019*)—by tracking the positions of user-defined body parts (*Mathis et al., 2018*; *Pereira et al., 2019*). These advances have drastically increased the quality and quantity, as well as the diversity, of behavioral data that are potentially available to researchers for answering scientific questions.

## Animal pose estimation using deep learning

In the past, conventional methods for measuring posture with computer vision relied on species-specific algorithms (*Uhlmann et al., 2017*), highly specialized or restrictive experimental setups (*Mendes et al., 2013*; *Kain et al., 2013*), attaching intrusive physical markers to the study animal (*Kain et al., 2013*), or some combination thereof. These methods also typically required expert computer-vision knowledge to use, were limited in the number or type of body parts that could be tracked (*Mendes et al., 2013*), involved capturing and handling the study animals to attach markers (*Kain et al., 2013*)—which is not possible for many species—and despite best efforts to minimize human involvement, often required manual intervention to correct errors (*Uhlmann et al., 2017*). These methods were all built to work for a small range of conditions and typically required considerable effort to adapt to novel contexts.

In contrast to conventional computer-vision methods, modern deep-learning– based methods can be used to achieve near human-level accuracy in almost any scenario by manually annotating data (*Figure 1*)—known as a *training set*—and training a general-purpose image-processing algorithm—a

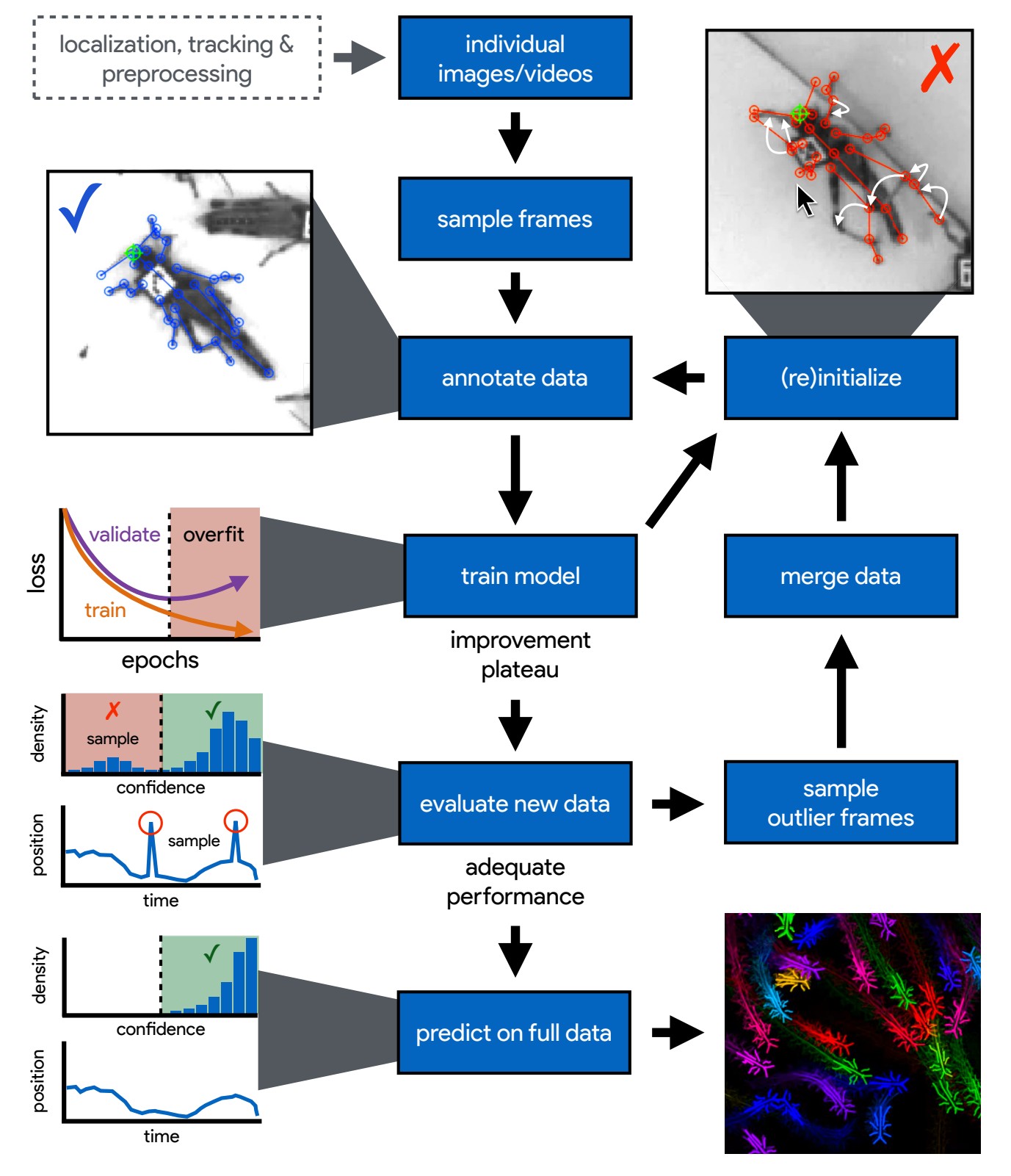

**Figure 1.** An illustration of the workflow for DeepPoseKit. Multi-individual images are localized, tracked, and preprocessed into individual images, which is not required for single-individual image datasets. An initial image set is sampled, annotated, and then iteratively updated using the active learning approach described by *Pereira et al. (2019)* (see Appendix 3). As annotations are made, the model is trained (*Figure 2*) with the current training set and keypoint locations are initialized for unannotated data to reduce the difficulty of further annotations. This is repeated until there is a

*Figure 1 continued on next page*

*Figure 1 continued*

noticeable improvement plateau for the initialized data—where the annotator is providing only minor corrections—and for the validation error when training the model (*Appendix 1—figure 4*). New data from the full dataset are evaluated with the model, and the training set is merged with new examples that are sampled based on the model's predictive performance, which can be assessed with techniques described by *Mathis et al. (2018)* and *Nath et al. (2019)* for identifying outlier frames and minimizing extreme prediction errors—shown here as the distribution of confidence scores predicted by the model and predicted body part positions with large temporal derivatives, indicating extreme errors. This process is repeated as necessary until performance is adequate when evaluating new data. The pose estimation model can then be used to make predictions for the full data set, and the data can be used for further analysis.

The online version of this article includes the following video for figure 1:

**Figure 1—video 1.** A visualization of the posture data output for a group of locusts (5× speed).

https://elifesciences.org/articles/47994#fig1video1

convolutional neural network or CNN—to automatically estimate the locations of an animal's body parts directly from images (*Figure 2*). State-of-the-art machine learning methods, like CNNs, use these training data to parameterize a model describing the statistical relationships between a set of input data (i.e., images) and the desired output distribution (i.e., posture keypoints). After adequate training, a model can be used to make predictions on previously-unseen data from the same data-set—inputs that were not part of the training set, which is known as *inference*. In other words, these models are able to generalize human-level expertise at scale after having been trained on only a

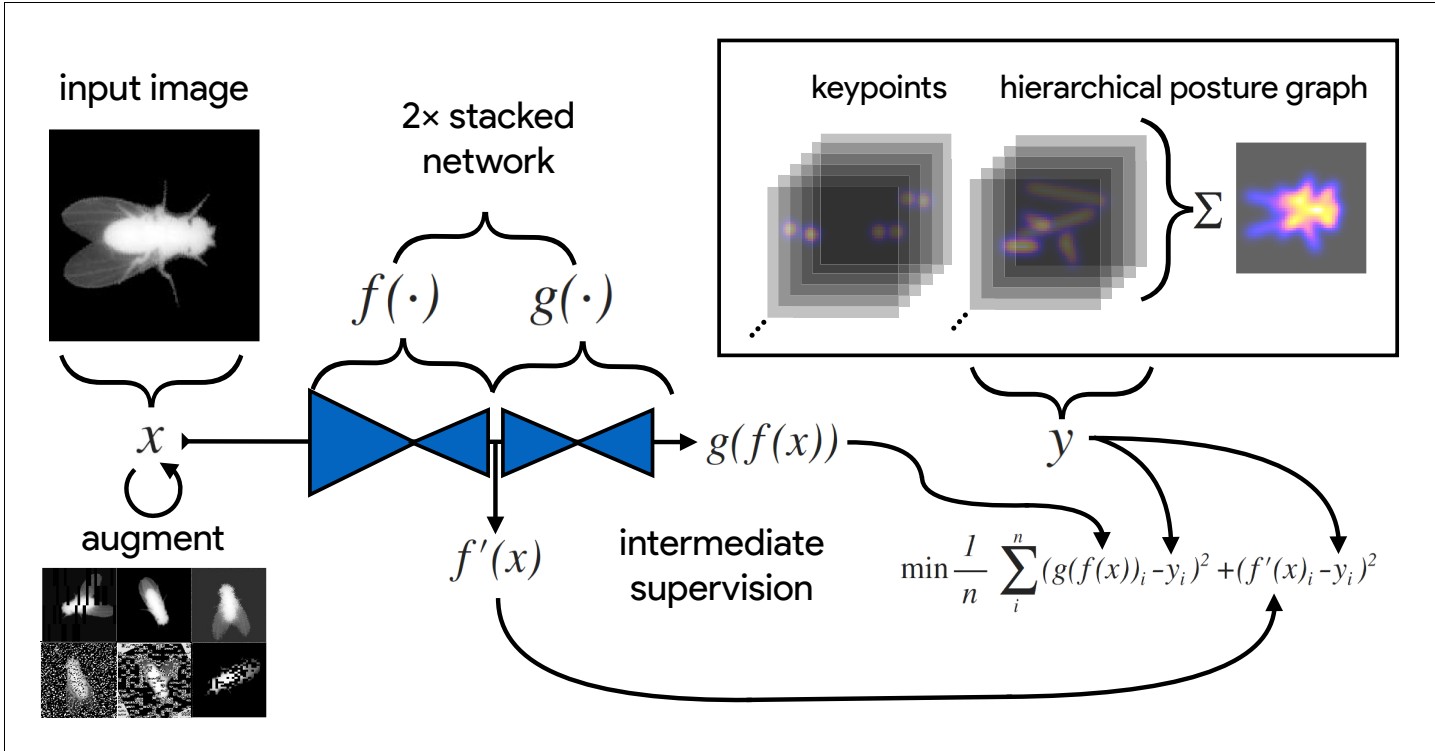

**Figure 2.** An illustration of the model training process for our Stacked DenseNet model in DeepPoseKit (see Appendix 2 for details about training models). Input images $x$ (top-left) are augmented (bottom-left) with various spatial transformations (rotation, translation, scale, etc.) followed by noise transformations (dropout, additive noise, blurring, contrast, etc.) to improve the robustness and generalization of the model. The ground truth annotations are then transformed with matching spatial augmentations (not shown for the sake of clarity) and used to draw the confidence maps $y$ for the keypoints and hierarchical posture graph (top-right). The images $x$ are then passed through the network to produce a multidimensional array $g(f(x))$—a stack of images corresponding to the keypoint and posture graph confidence maps for the ground truth $y$. Mean squared error between the outputs for both networks $g(f(x))$ and $f'(x)$ and the ground truth data $y$ is then minimized (bottom-right), where $f'(x)$ indicates a subset of the output from $f(x)$—only those feature maps being optimized to reproduce the confidence maps for the purpose of intermediate supervision (Appendix 5). The loss function is minimized until the validation loss stops improving—indicating that the model has converged or is starting to overfit to the training data.

relatively small number of examples. We provide more detailed background information on using CNNs for pose estimation in Appendices 2–6.

Similar to conventional pose estimation methods, the task of implementing deep-learning models in software and training them on new data is complex and requires expert knowledge. However, in most cases, once the underlying model and training routine are implemented, a high-accuracy pose estimation model for a novel context can be built with minimal modification—often just by changing the training data. With a simplified toolkit and high-level software interface designed by an expert, even scientists with limited computer-vision knowledge can begin to apply these methods to their research. Once the barriers for implementing and training a model are sufficiently reduced, the main bottleneck for using these methods becomes collecting an adequate training set—a labor-intensive task made less time-consuming by techniques described in Appendix 3.

*Mathis et al. (2018)* and *Pereira et al. (2019)* were the first to popularize the use of CNNs for animal pose estimation. These researchers built on work from the human pose estimation literature (e.g., *Andriluka et al., 2014*; *Insafutdinov et al., 2016*; *Newell et al., 2016*) using a type of *fully-convolutional neural network* or *F-CNN* (*Long et al., 2015*; Appendix 4) often referred to as an *encoder-decoder* model (Appendix 4: 'Encoder-decoder models'). These models are used to measure animal posture by training the network to transform images into probabilistic estimates of key-point locations, known as *confidence maps* (shown in *Figure 2*), that describe the body posture for one or more individuals. These confidence maps are processed to produce the 2-D spatial coordinates of each keypoint, which can then be used for further analysis.

While deep-learning models typically need large amounts of training data, both *Mathis et al. (2018)* and *Pereira et al. (2019)* have demonstrated that near human-level accuracy can be achieved with few training examples (Appendix 3). In order to ensure generalization to large datasets, both groups of researchers introduced ideas related to iteratively refining the training set used for model fitting (*Mathis et al., 2018*; *Pereira et al., 2019*). In particular, *Pereira et al. (2019)* describe a technique known as *active learning* where a trained model is used to initialize new training data and reduce annotation time (Appendix 3). *Mathis et al. (2018)* describe multiple techniques that can be used to further refine training data and minimize errors when making predictions on the full dataset. Simple methods to accomplish this include filtering data or selecting new training examples based on confidence scores or the entropy of the confidence maps from the model output. *Nath et al. (2019)* also introduced the use temporal derivatives (i.e., speed and acceleration) and autoregressive models to identify outlier frames, which can then be labeled to refine the training set or excluded from further analysis on the final dataset (*Figure 1*).

## Pose estimation models and the speed-accuracy trade-off

*Mathis et al., 2018* developed their pose estimation model, which they call *DeepLabCut*, by modifying a previously published model called *DeeperCut* (*Insafutdinov et al., 2016*). The DeepLabCut model (*Mathis et al., 2018*), like the DeeperCut model, is built on the popular *ResNet* architecture (*He et al., 2016*)—a state-of-the-art deep-learning model used for image classification. This choice is advantageous because the use of a popular architecture allows for incorporating a pre-trained encoder to improve performance and reduce the number of required training examples (*Mathis et al., 2018*), known as *transfer learning* (*Pratt, 1992*; Appendix 3)—although, as will be seen, transfer learning appears to offer little improvement over a randomly initialized model. However, this choice of of a pre-trained architecture is also disadvantageous as the model is *overparameterized* with >25 million parameters. Overparameterization allows the model to make accurate predictions, but this may come with the cost of slow inference. To alleviate these effects, work from *Mathis and Warren (2018)* showed that inference speed for the DeepLabCut model (*Mathis et al., 2018*) can be improved by decreasing the resolution of input images, but this is achieved at the expense of accuracy.

With regard to model design, *Pereira et al. (2019)* implement a modified version of a model called *SegNet* (*Badrinarayanan et al., 2015*), which they call *LEAP* (LEAP Estimates Animal Pose), that attempts to limit model complexity and overparameterization with the goal of maximizing inference speed (see Appendix 6)—however, our comparisons from this paper suggest (*Pereira et al., 2019*) achieved only limited success compared to the DeepLabCut model (*Mathis et al., 2018*). The LEAP model is advantageous because it is explicitly designed for fast inference but has disadvantages such as a lack of robustness to data variance, like rotations or shifts in lighting, and an inability

to generalize to new experimental setups. Additionally, to achieve maximum performance, the training routine for the LEAP model introduced by *Pereira et al. (2019)* requires computationally expensive preprocessing that is not practical for many datasets, which makes it unsuitable for a wide range of experiments (see Appendix 6 for more details).

Together the methods from *Mathis et al. (2018)* and *Pereira et al. (2019)* represent the two extremes of a phenomenon known as the *speed-accuracy trade-off* (*Huang et al., 2017b*)—an active area of research in the machine learning literature. *Mathis et al. (2018)* prioritize accuracy over speed by using a large overparameterized model (*Insafutdinov et al., 2016*), and *Pereira et al. (2019)* prioritize speed over accuracy by using a smaller less-robust model. While this speed-accuracy trade-off can limit the capabilities of CNNs, there has been extensive work to make these models more efficient without impacting accuracy (e.g., *Chollet, 2017*; *Huang et al., 2017a*; *Sandler et al., 2018*). To address the limitations of this trade-off, we apply recent developments from the machine learning literature and provide an effective solution to the problem.

In the case of F-CNN models used for pose estimation, improvements in efficiency and robustness have been made through the use of *multi-scale inference* (Appendix 4: 'Encoder-decoder models') by increasing connectivity between the model's many layers across multiple spatial scales (*Appendix 4—figure 1*) Multi-scale inference implicitly allows the model to simultaneously integrate large-scale global information, such as the lighting, image background, or the orientation of the focal individual's body trunk; information from intermediate scales like anatomical geometry related to cephalization and bilateral symmetry; and fine-scale local information that could include differences in color, texture, or skin patterning for specific body parts. This multi-scale design gives the model capacity to learn the hierarchical relationships between different spatial scales and efficiently aggregate them into a joint representation when solving the posture estimation task (see Appendix 4: 'Encoder-decoder models' and *Appendix 4—figure 1* for further discussion).

## Individual vs. multiple pose estimation

Most work on human pose estimation now focuses on estimating the pose of multiple individuals in an image (e.g. *Cao et al., 2017*). For animal pose estimation, the methods from *Pereira et al. (2019)* are limited to estimating posture for single individuals—known as *individual pose estimation*—while the methods from *Mathis et al. (2018)* can also be extended to estimate posture for multiple individuals simultaneously—known as *multiple pose estimation*. However, the majority of work on multiple pose estimation, including *Mathis et al. (2018)*, has not adequately solved the tracking problem of linking individual posture data across frames in a video, especially after visual occlusions, which are common in many behavioral experiments—although recent work has attempted to address this problem (*Iqbal et al., 2017*; *Andriluka et al., 2018*). Additionally, as the name suggests, the task of multiple pose estimation requires exhaustively annotating images of multiple individuals—where every individual in the image must be annotated to prevent the model from learning conflicting information. This type of annotation task is even more laborious and time consuming than annotations for individual pose estimation and the amount of labor increases proportionally with the number of individuals in each frame, which makes this approach intractable for many experimental systems.

Reliably tracking the position of individuals over time is important for most behavioral studies, and there are a number of diverse methods already available for solving this problem (*Pérez-Escudero et al., 2014*; *Crall et al., 2015*; *Graving, 2017*; *Romero-Ferrero et al., 2019*; *Wild et al., 2018*; *Boenisch et al., 2018*). Therefore, to avoid solving an already-solved problem of tracking individuals and to circumvent the cognitively complex task of annotating data for multiple pose estimation, the work we describe in this paper is purposefully limited to individual pose estimation—where each image contains only a single focal individual, which may be cropped from a larger multi-individual image after localization and tracking. We introduce a top-down posture estimation framework that can be readily adapted to existing behavioral analysis workflows, which could include any method for localizing and tracking individuals.

The additional step of localizing and tracking individuals naturally increases the processing time for producing posture data from raw image data, which varies depending on the algorithms being used and the number of individuals in each frame. While tracking and localization may not be practical for all experimental systems, which could make our methods difficult to apply 'out-of-the-box', the increased processing time from automated tracking algorithms is a reasonable trade-off for most

systems given the costly alternative of increased manual labor when annotating data. This trade-off seems especially practical when considering that the posture data produced by most multiple pose estimation algorithms still need to be linked across video frames to maintain the identity of each individual, which is effectively a bottom-up method for achieving the same result. Limiting our methods to individual pose estimation also simplifies the pose detection problem as processing confidence maps produced by the model does not require computationally-expensive local peak detection and complex methods for grouping keypoints into individual posture graphs (e.g. *Insafutdinov et al., 2016*; *Cao et al., 2017*; Appendix 4). Additionally, because individual pose estimation is such a well-studied problem in computer vision, we can readily build on state-of-the-art methods for this task (see Appendices 4 and 5 for details).

## Results

Here, we introduce fast, flexible, and robust pose estimation methods, with a software interface—a high-level programming interface (API) and graphical user-interface (GUI) for annotations—that emphasizes usability. Our methods build on the state-of-the-art for individual pose estimation (*Newell et al., 2016*; Appendix 5), convolutional regression models (*Jégou et al., 2017*; Appendix 4: 'Encoder-decoder models'), and conventional computer vision algorithms (*Guizar-Sicairos et al., 2008*) to improve model efficiency and achieve faster, more accurate results on multiple challenging pose estimation tasks. We developed two model implementations—including a new model architecture that we call *Stacked DenseNet*—and a new method for processing confidence maps called *sub-pixel maxima* that provides fast and accurate peak detection for estimating keypoint locations with subpixel precision—even at low spatial resolutions. We also discuss a modification to incorporate a hierarchical posture graph for learning the multi-scale geometry between keypoints on the animal's body, which increases accuracy when training pose estimation models. We ran experiments to optimize our approach and compared our new models to the models from *Mathis et al. (2018)* (Deep-LabCut) and *Pereira et al. (2019)* (LEAP) in terms of speed, accuracy, training time, and generalization ability. We benchmarked these models using three image datasets recorded in the laboratory and the field—including multiple interacting individuals that were first localized and cropped from larger, multi-individual images (see 'Materials and methods' for details).

### An end-to-end pose estimation framework

We provide a full-featured, extensible, and easy-to-use software package that is written entirely in the Python programming language (Python Software Foundation) and is built using TensorFlow as a backend (*Abadi et al., 2015*). Our software is a complete, end-to-end pipeline (*Figure 1*) with a custom GUI for creating annotated training data with active learning similar to *Pereira et al. (2019)* (Appendix 3), as well as a flexible pipeline for data augmentation (*Jung, 2018*; Appendix 3; shown in *Figure 2*), model training and evaluation (*Figure 2*; Appendix 2), and running inference on new data. We designed our high-level programming interface using the same guidelines from Keras (*keras team, 2015*) to allow the user to go from idea to result as quickly as possible, and we organized our software into a Python module called *DeepPoseKit*. The code, documentation, and examples for our entire software package are freely available at https://github.com/jgraving/deepposekit under a permissive open-source license.

### Our pose estimation models

To achieve the goal of 'fast animal pose estimation' introduced by *Pereira et al. (2019)*, while maintaining the robust predictive power of models like DeepLabCut (*Mathis et al., 2018*), we implemented two fast pose estimation models that extend the state-of-the-art model for individual pose estimation introduced by *Newell et al. (2016)* and the current state-of-the art for convolutional regression from *Jégou et al. (2017)*. Our model implementations use fewer parameters than both the DeepLabCut model (*Mathis et al., 2018*) and LEAP model (*Pereira et al., 2019*) while simultaneously removing many of the limitations of these architectures.

In order to limit overparameterization while minimizing performance loss, we designed our models to allow for multi-scale inference (Appendix 4: 'Encoder-decoder models') while optimizing our model hyperparameters for efficiency. Our first model is a novel implementation of *FC-DenseNet* from *Jégou et al. (2017)* (Appendix 4: 'Encoder-decoder models') arranged in a stacked

configuration similar to *Newell et al. (2016)* (Appendix 5). We call this new model Stacked Dense-Net, and to the best of our knowledge, this is the first implementation of this model architecture in the literature—for pose estimation or otherwise. Further details for this model are available in Appendix 8. Our second model is a modified version of the *Stacked Hourglass* model from *Newell et al. (2016)* (Appendix 5) with hyperparameters that allow for changing the number of filters in each convolutional block to constrain the number of parameters—rather than using 256 filters for all layers as described in *Newell et al. (2016)*.

## Subpixel keypoint prediction on the GPU allows for fast and accurate inference

In addition to implementing our efficient pose estimation models, we developed a new method to process model outputs to allow for faster, more accurate predictions. When using a fully-convolutional posture estimation model, the confidence maps produced by the model must be converted into coordinate values for the predictions to be useful, and there are typically two choices for making this conversion. The first is to move the confidence maps out of GPU memory and post-process them on the CPU. This solution allows for easy, flexible, and accurate calculation of the coordinates with subpixel precision (*Insafutdinov et al., 2016*; *Mathis et al., 2018*). However, CPU processing is not ideal because moving large arrays of data between the GPU and CPU can be costly, and computation on the CPU is generally slower. The other option is to directly process the confidence maps on the GPU and then move the coordinate values from the GPU to the CPU. This approach usually means converting confidence maps to integer coordinates based on the row and column index of the global maximum for each confidence map (*Pereira et al., 2019*). However, this means that, to achieve a precise estimation, the confidence maps should be predicted at the full resolution of the input image, or larger, which slows down inference speed.

As an alternative to these two strategies, we introduce a new GPU-based convolutional layer that we call *subpixel maxima*. This layer uses the fast, efficient, image registration algorithm introduced by *Guizar-Sicairos et al. (2008)* to translationally align a two-dimensional Gaussian filter to each confidence map via Fourier-based convolution. The translational shift between the filter and each confidence map allows us to calculate the coordinates of the global maxima with high-speed and subpixel precision. This technique allows for accurate predictions of keypoint locations even if the model's confidence maps are dramatically smaller than the resolution of the input image. We compared the accuracy of our subpixel maxima layer to an integer-based maxima layer using the fly dataset from *Pereira et al. (2019)* (see 'Materials and methods'). We found significant accuracy improvements across every downsampling configuration (*Appendix 1—figure 1a*). Even with confidence maps at $\frac{1}{8}\times$ the resolution of the original image, error did not drastically increase compared to full-resolution predictions. Making predictions for confidence maps at such a downsampled resolution allows us to achieve very fast inference >1000 Hz while maintaining high accuracy (*Appendix 1—figure 1b*).

We also provide speed comparisons with the other models we tested and find that our Stacked DenseNet model with our subpixel peak detection algorithm is faster than the DeepLabCut model (*Mathis et al., 2018*) for both offline (batch size = 100) and real-time speeds (batch size = 1). While we find that our Stacked DenseNet model is faster than the LEAP model (*Pereira et al., 2019*) for offline processing (batch size = 100), the LEAP model (*Pereira et al., 2019*) is significantly faster for real-time processing (batch size = 1). Our Stacked Hourglass model (*Newell et al., 2016*) is about the same or slightly faster than Stacked DenseNet for offline speeds (batch size = 100), but is much slower for real-time processing (batch size = 1). Achieving fast pose estimation using CNNs typically relies on massively parallel processing on the GPU with large batches of data or requires downsampling the images to increase speed, which increases error (*Mathis and Warren, 2018*). These factors make fast and accurate real-time inference challenging to accomplish. Our Stacked DenseNet model, with a batch size of one, can run inference at ~30–110 Hz—depending on the resolution of the predicted confidence maps (*Appendix 1—figure 1b*). These speeds are faster than the DeepLabCut model (*Mathis et al., 2018*) and could be further improved by downsampling the input image resolution or reconfiguring the model with fewer parameters. This allows our methods to be flexibly used for real-time or closed-loop behavioral experiments with prediction errors similar to current state-of-the-art methods.

## Learning multi-scale geometry between keypoints improves accuracy and reduces extreme errors

Minimizing extreme prediction errors is important to prevent downstream effects on any further behavioral analysis (*Seethapathi et al., 2019*)—especially in the case of analyses based on time-frequency transforms like those from *Berman et al. (2014b)*, *Berman et al. (2016)*, *Klibaite et al. (2017)*, *Todd et al. (2017)*, *Klibaite and Shaevitz (2019)* and *Pereira et al. (2019)* where high magnitude errors can cause inaccurate behavioral classifications. While effects of these extreme errors can be minimized using post-hoc filters and smoothing, these post-processing techniques can remove relevant high-frequency information from time-series data, so this solution is less than ideal. One way to minimize extreme errors when estimating posture is to incorporate multiple spatial scales when making predictions (e.g., *Chen et al., 2017*). Our pose estimation models are implicitly capable of using information from multiple scales (see Appendix 4: 'Encoder-decoder models'), but there is no explicit signal that optimizes the model to take advantage of this information when making predictions.

To remedy this, we modified the model's output to predict, in addition to keypoint locations, a hierarchical graph of edges describing the multi-scale geometry between keypoints—similar to the part affinity fields described by *Cao et al. (2017)*. This was achieved by adding an extra set of confidence maps to the output where edges in the postural graph are represented by Gaussian-blurred lines the same width as the Gaussian peaks in the keypoint confidence maps. Our posture graph output then consists of four levels: (1) a set of confidence maps for the smallest limb segments in the graph (e.g. foot to ankle, knee to hip, etc.; *Figure 2*), (2) a set of confidence maps for individual limbs (e.g. left leg, right arm, etc.; Figure 4), (3) a map with the entire postural graph, and (4) a fully integrated map that incorporates the entire posture graph and confidence peaks for all of the joint locations (*Figure 2*). Each level of the hierarchical graph is built from lower levels in the output, which forces the model to learn correlated features across multiple scales when making predictions.

We find that training our Stacked DenseNet model to predict a hierarchical posture graph reduces keypoint prediction error (*Appendix 1—figure 2*), and because the feature maps for the posture graph can be removed from the final output during inference, this effectively improves prediction accuracy for free. Both the mean and variance of the error distributions were lower when predicting the posture graph, which suggests that learning multi-scale geometry both decreases error on average and helps to reduce extreme prediction errors. The overall effect size for this decrease in error is fairly small (<1 pixel average reduction in error), but based on the results from the zebra dataset, this modification more dramatically improves performance for datasets with higher variance images and sparse posture graphs. Predicting the posture graph may be especially useful for animals with long slender appendages such as insect legs and antennae where prediction errors are likely to occur due to occlusions and natural variation in the movement of these body parts. These results also suggest that annotating multiple keypoints to incorporate an explicit signal for multi-scale information may help improve prediction accuracy for a specific body part of interest.

## Stacked DenseNet is fast and robust

We benchmarked our new model implementations against the models *Pereira et al. (2019)* and *Mathis et al. (2018)*. We find that our Stacked DenseNet model outperforms both the LEAP model (*Pereira et al., 2019*) and the DeepLabCut model (*Mathis et al., 2018*) in terms of speed while also achieving much higher accuracy than the LEAP model (*Pereira et al., 2019*) with similar accuracy to the DeepLabCut model (*Mathis et al., 2018*; *Figure 3a*). We found that both the Stacked Hourglass and Stacked DenseNet models outperformed the LEAP model (*Pereira et al., 2019*). Notably our Stacked DenseNet model achieved approximately 2× faster inference speeds with 3× higher mean accuracy. Not only were our models average prediction error significantly improved, but also, importantly, the variance was lower—indicating that our models produced fewer extreme prediction errors. At $\frac{1}{4}\times$ resolution, our Stacked DenseNet model consistently achieved prediction accuracy nearly identical to the DeepLabCut model (*Mathis et al., 2018*) while running inference at nearly 2× the speed and using only ~5% of the parameters—1.5 million vs. ~26 million. Detailed results of our model comparisons are shown in *Figure 3—figure supplement 1*.

While the Stacked DenseNet model used for comparisons is already fast, inference speed could be further improved by using a $\frac{1}{8}\times$ output without much increase in error (*Appendix 1—figure 1*) or

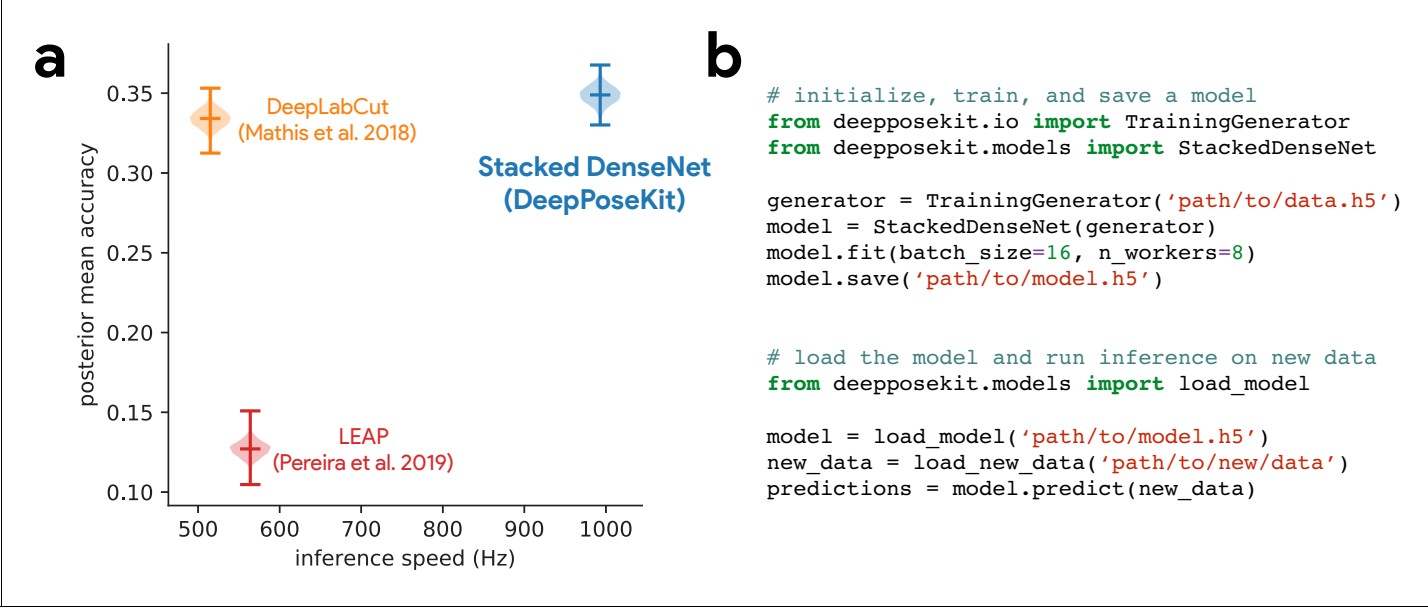

**Figure 3.** DeepPoseKit is fast, accurate, and easy-to-use. Our Stacked DenseNet model estimates posture at approximately 2×—or greater—the speed of the LEAP model (*Pereira et al., 2019*) and the DeepLabCut model (*Mathis et al., 2018*) while also achieving similar accuracy to the DeepLabCut model (*Mathis et al., 2018*)—shown here as mean accuracy $(1 + Euclidean error)^{-1}$ for our most challenging dataset of multiple interacting Grévy's zebras (*E. grevyi*) recorded in the wild (a). See *Figure 3—figure supplement 1* for further details. Our software interface is designed to be straightforward but flexible. We include many options for expert users to customize model training with sensible default settings to make pose estimation as easy as possible for beginners. For example, training a model and running inference on new data requires writing only a few lines of code and specifying some basic settings (b).

The online version of this article includes the following figure supplement(s) for figure 3:

**Figure supplement 1.** Euclidean error distributions for each model across our three datasets.

by further adjusting the hyperparameters to constrain the size of the model. Our Stacked Hourglass implementation followed closely behind the performance of our Stacked DenseNet model and the DeepLabCut model (*Mathis et al., 2018*) but consistently performed more poorly than our Stacked DenseNet model in terms of prediction accuracy, so we excluded this model from further analysis. We were also able to reproduce the results reported by *Pereira et al. (2019)* that the LEAP model and the Stacked Hourglass model (*Newell et al., 2016*) have similar average prediction error for the fly dataset. However, we also find that the LEAP model (*Pereira et al., 2019*) has much higher variance, which suggests it is more prone to extreme prediction errors—a problem for further data analysis.

## Stacked DenseNet trains quickly and requires few training examples

To further compare models, we used our zebra dataset to assess the training time needed for our Stacked DenseNet model, the DeepLabCut model (*Mathis et al., 2018*), and the LEAP model (*Pereira et al., 2019*) to reach convergence (i.e., complete training) as well as the amount of training data needed for each model to generalize to new data from outside the training set. We find that our Stacked DenseNet model, the DeepLabCut model (*Mathis et al., 2018*), and the LEAP model (*Pereira et al., 2019*) all fully converge in just a few hours and reach reasonably high accuracy after only an hour of training (*Appendix 1—figure 3*). However, it appears that our Stacked DenseNet model tends to converge to a good minimum faster than both the DeepLabCut model (*Mathis et al., 2018*) and the LEAP model (*Pereira et al., 2019*).

We also show that our Stacked DenseNet model achieves good generalization with few training examples and without the use of transfer learning (*Appendix 1—figure 4*). These results demonstrate that, when combined with data augmentation, as few as five training examples can be used as an initial training set for labelling keypoints with active learning (*Figure 1*). Additionally, because our analysis shows that generalization to new data plateaus after approximately 100 labeled training

examples, it appears that 100 training examples is a reasonable minimum size for a training set—although the exact number will likely change depending the variance of the image data being annotated. To further examine the effect of transfer learning on model generalization, we compared performance between the DeepLabCut model (*Mathis et al., 2018*) initialized with weights pretrained on the ImageNet database (*Deng et al., 2009*) vs. the same model with randomly initialized weights (*Appendix 1—figure 4*). As postulated by *Mathis et al. (2018)*, we find that transfer learning does provide some benefit to the DeepLabCut model's ability to generalize. However, the effect size of this improvement is small with a mean reduction in Euclidean error of <0.5 pixel. Together these results indicate that transfer learning is helpful, but not required, for deep learning models to achieve good generalization with limited training data.

## Discussion

Here, we have presented a new software toolkit, called DeepPoseKit, for estimating animal posture using deep learning models. We built on the state-of-the-art for individual pose estimation using convolutional neural networks to achieve fast inference without reducing accuracy or generalization ability. Our new pose estimation model, called Stacked DenseNet, offers considerable improvements (*Figure 3a*; *Figure 3—figure supplement 1*) over the models from *Mathis et al. (2018)* (DeepLabCut) and *Pereira et al. (2019)* (LEAP), and our software framework also provides a simplified interface (*Figure 3b*) for using these advanced tools to measure animal behavior and locomotion. We tested our methods across a range of datasets from controlled laboratory environments with single individuals to challenging field situations with multiple interacting individuals and variable lighting conditions. We found that our methods perform well for all these situations and require few training examples to achieve good predictive performance on new data—without the use of transfer learning. We ran experiments to optimize our approach and discovered that some straightforward modifications can greatly improve speed and accuracy. Additionally, we demonstrated that these modifications improve not the just the average error but also help to reduce extreme prediction errors—a key determinant for the reliability of subsequent statistical analysis.

While our results offer a good-faith comparison of the available methods for animal pose estimation, there is inherent uncertainty that we have attempted to account for but may still bias our conclusions. For example, deep learning models are trained using stochastic optimization algorithms that give different results with each replicate, and the Bayesian statistical methods we use for comparison are explicitly probabilistic in nature. There is also great variability across hardware and software configurations when using these models in practice (*Mathis and Warren, 2018*), so performance may change across experimental setups and datasets. Additionally, we demonstrated that some models may perform better than others for specific applications (*Figure 3—figure supplement 1*), and to account for this, our toolkit offers researchers the ability to choose the model that best suits their requirements—including the LEAP model (*Pereira et al., 2019*) and the DeepLabCut model (*Mathis et al., 2018*).

We highlighted important considerations when using CNNs for pose estimation and reviewed the progress of fully convolutional regression models from the literature. The latest advancements for these models have been driven mostly by a strategy of adding more connections between layers to increase performance and efficiency (e.g., *Jégou et al., 2017*). Future progress for this class of models may require better loss functions (*Goodfellow et al., 2014*; *Johnson et al., 2016a*; *Chen et al., 2017*; *Zhang et al., 2018*), models that more explicitly incorporate the spatial dependencies within a scene (*Van den Oord et al., 2016b*), and temporal structure of the data (*Seethapathi et al., 2019*), as well as more mathematically principled approaches (e.g., *Weigert et al., 2018*; *Roy et al., 2019*) such as the application of formal probabilistic concepts (*Kendall and Gal, 2017*) and Bayesian inference at scale (*Tran et al., 2018*).

Measuring behavior is a critical factor for many studies in neuroscience (*Krakauer et al., 2017*). Understanding the connections between brain activity and behavioral output requires detailed and objective descriptions of body posture that match the richness and resolution neural measurement technologies have provided for years (*Anderson and Perona, 2014*; *Berman, 2018*; *Brown and de Bivort, 2018*), which our methods and other deep-learning– based tools provide (*Mathis et al., 2018*; *Pereira et al., 2019*). We have also demonstrated the possibility that our toolkit could be used for real-time inference, which allows for closed-loop experiments where sensory stimuli or

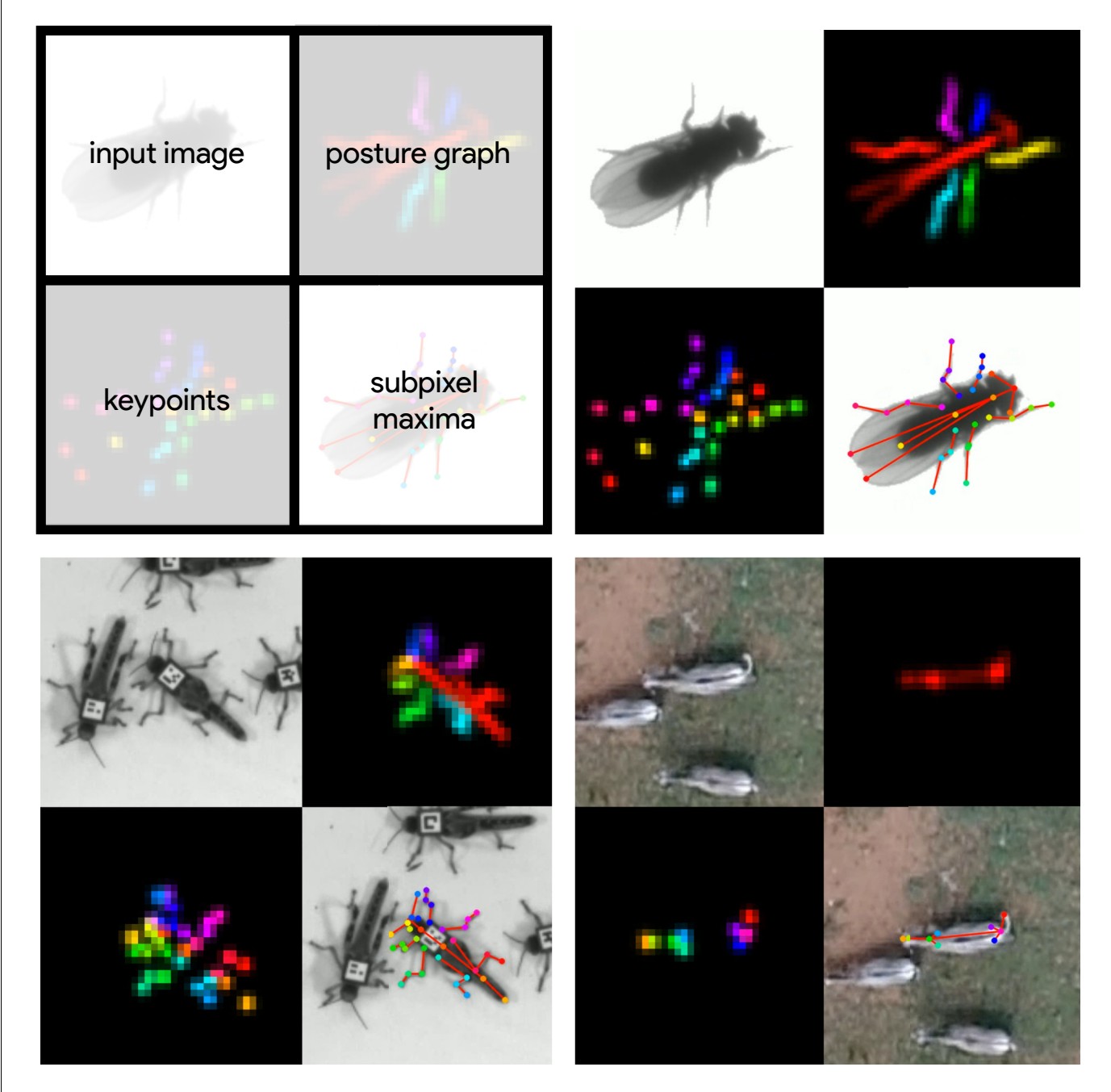

**Figure 4.** Datasets used for evaluation. A visualization of the datasets we used to evaluate our methods (*Table 1*). For each dataset, confidence maps for the keypoints (bottom-left) and posture graph (top-right) are illustrated using different colors for each map. These outputs are from our Stacked DenseNet model at $\frac{1}{4}\times$ resolution.

The online version of this article includes the following video(s) for figure 4:

**Figure 4—video 1.** A video of a behaving fly from *Pereira et al. (2019)* with pose estimation outputs visualized.
https://elifesciences.org/articles/47994#fig4video1
**Figure 4—video 2.** A video of a behaving locust with pose estimation outputs visualized.
https://elifesciences.org/articles/47994#fig4video2
**Figure 4—video 3.** A video of a behaving Grévy's zebra with pose estimation outputs visualized.
https://elifesciences.org/articles/47994#fig4video3

optogenetic stimulation are controlled in response to behavioral measurements (e.g., *Bath et al., 2014*; *Stowers et al., 2017*). Using real-time measurements in conjunction with optogenetics or thermogenetics may be key to disentangling the causal structure of motor output from the brain—especially given that recent work has shown an animal's response to optogenetic stimulation can differ depending on the behavior it is currently performing (*Cande et al., 2018*). Real-time behavioral quantification is also particularly important as closed-loop virtual reality is quickly becoming an indispensable tool for studying sensorimotor relationships in individuals and collectives (*Stowers et al., 2017*).

Quantifying individual movement is essential for revealing the genetic (*Kain et al., 2012*; *Brown et al., 2013*; *Ayroles et al., 2015*) and environmental (*Bierbach et al., 2017*; *Akhund-Zade et al., 2019*; *Versace et al., 2019*) underpinnings of phenotypic variation in behavior—as well as the phylogeny of behavior (e.g., *Berman et al., 2014a*). Measuring individual behavioral phenotypes requires tools that are robust, scaleable, and easy-to-use, and our approach offers the ability to quickly and accurately quantify the behavior of many individuals in great detail. When combined with tools for genetic manipulations (*Ran et al., 2013*; *Doudna and Charpentier, 2014*), high-throughput behavioral experiments (*Alisch et al., 2018*; *Javer et al., 2018*; *Werkhoven et al., 2019*), and behavioral analysis (e.g., *Berman et al., 2014b*; *Wiltschko et al., 2015*), our methods could help to provide the data resolution and statistical power needed for dissecting the complex relationships between genes, environment, and behavioral variation.

When used together with other tools for localization and tracking (e.g., *Pérez-Escudero et al., 2014*; *Crall et al., 2015*; *Graving, 2017*; *Romero-Ferrero et al., 2019*; *Wild et al., 2018*; *Boenisch et al., 2018*), our methods are capable of reliably measuring posture for multiple interacting individuals. The importance of measuring detailed representations of individual behavior when studying animal collectives has been well established (*Strandburg-Peshkin et al., 2013*; *Rosenthal et al., 2015*; *Strandburg-Peshkin et al., 2015*; *Strandburg-Peshkin et al., 2017*). Estimating body posture is an essential first step for unraveling the sensory networks that drive group coordination, such as vision-based networks measured via raycasting (*Strandburg-Peshkin et al., 2013*; *Rosenthal et al., 2015*). Additionally, using body pose estimation in combination with computational models of behavior (e.g., *Costa et al., 2019*; *Wiltschko et al., 2015*) and unsupervised behavioral classification methods (e.g., *Berman et al., 2014b*; *Pereira et al., 2019*) may allow for further dissection of how information flows through groups by revealing the networks of behavioral contagion across multiple timescales and sensory modalities. While we have provided a straightforward solution for applying existing pose estimation methods to measure collective behavior, there still remain many challenging scenarios where these methods would fail. For example, tracking posture in a densely packed bee hive or school of fish would require novel solutions to deal with the 3-D nature of individual movement, which includes maintaining individual identities and dealing with the resulting occlusions that go along with imaging these types of biological systems.

When combined with unmanned aerial vehicles (UAVs; *Schiffman, 2014*) or other field-based imaging (*Francisco et al., 2019*), applying these methods to the study of individuals and groups in the wild can provide high-resolution behavioral data that goes beyond the capabilities of current GPS and accelerometry-based technologies (*Nagy et al., 2010*; *Nagy et al., 2013*; *Kays et al., 2015*; *Strandburg-Peshkin et al., 2015*; *Strandburg-Peshkin et al., 2017*; *Flack et al., 2018*)—especially for species that are impractical to study with tags or collars. Additionally, by applying these methods in conjunction with 3-D habitat reconstruction—using techniques from photogrammetry (*Strandburg-Peshkin et al., 2017*; *Francisco et al., 2019*)—field-based studies can begin to integrate fine-scale behavioral measurements with the full 3-D environment in which the behavior evolved. Future advances will likely allow for the calibration and synchronizaton of imaging devices across multiple UAVs (e.g., *Price et al., 2018*; *Saini et al., 2019*). This would make it possible to measure the full 3-D posture of wild animals (e.g., *Zuffi et al., 2019*) in scenarios where fixed camera systems (e.g., *Nath et al., 2019*) would not be tractable, such as during migratory or predation events. When combined, these technologies could allow researchers to address questions about the behavioral ecology of animals that were previously impossible to answer.

Computer vision algorithms for measuring behavior at the scale of posture have rapidly advanced in a very short time; nevertheless, the task of pose estimation is far from solved. There are hard limitations to this current generation of pose estimation methods that are primarily related to the requirement for human annotations and user-defined keypoints—both in terms of the number of

keypoints, the specific body parts being tracked, and the inherent difficulty of incorporating temporal information into the annotation and training procedures. Often the body parts chosen for annotation are an obvious fit for the experimental design and have reliably visible reference points on the animal's body that make them easy to annotate. However, in many cases the required number and type of body parts needed for data analysis may not be so obvious—such as in the case of unsupervised behavior classification methods (*Berman et al., 2014b*; *Pereira et al., 2019*). Additionally, the reference points for labeling images with keypoints can be hard to define and consistently annotate across images, which is often the case for soft or flexible-bodied animals like worms and fish. Moreover, due to the laborious nature of annotating keypoints, the current generation of methods also rarely takes into account the natural temporal structure of the data, instead treating each video frame as a statistically independent event, which can lead to extreme prediction errors (reviewed by *Seethapathi et al., 2019*). Extending these methods to track the full three-dimensional posture of animals also typically requires the use of multiple synchronized cameras (*Nath et al., 2019*; *Günel et al., 2019*), which increases the cost and complexity of creating an experimental setup, as well as the manual labor required for annotating a training set, which must include labeled data from every camera view.

These limitations make it clear that fundamentally-different methods may be required to move the field forward. New pose estimation methods are already replacing human annotations with fully articulated volumetric 3-D models of the animal's body (e.g., the SMAL model from *Zuffi et al., 2017* or the SMALST model from *Zuffi et al., 2019*), and the 3-D scene can be estimated using unsupervised, semi-supervised, or weakly-supervised methods (e.g., *Jaques et al., 2019*; *Zuffi et al., 2019*), where the shape, position, and posture of the animal's body, the camera position and lens parameters, and the background environment and lighting conditions are jointly learned directly from 2-D images by a deep-learning model (*Valentin et al., 2019*; *Zuffi et al., 2019*). These *inverse graphics models* (*Kulkarni et al., 2015*; *Sabour et al., 2017*; *Valentin et al., 2019*) take advantage of recently developed differentiable graphics engines that allow 3-D rendering parameters to be controlled using standard optimization methods (*Zuffi et al., 2019*; *Valentin et al., 2019*). After optimization, the volumetric 3-D timeseries data predicted by the deep learning model could be used directly for behavioral analysis or specific keypoints or body parts could be selected for analysis post-hoc. In order to more explicitly incorporate the natural statistical properties of the data, these models also apply perceptual loss functions (*Johnson et al., 2016a*; *Zhang et al., 2018*; *Zuffi et al., 2019*) and could be extended to use adversarial (*Goodfellow et al., 2014*; *Chen et al., 2017*) loss functions, both of which incorporate spatial dependencies within the scene rather than modelling each video frame as a set of statistically independent pixel distributions—as is the case with current methods that use likelihood functions such as pixel-wise mean squared error (e.g., *Pereira et al., 2019*) or cross-entropy loss (e.g., *Mathis et al., 2018*). Because there is limited or no requirement for human-provided labels with these new methods, these models could also be easily modified to incorporate the temporal structure of the data using autoregressive representations (e.g., *Van den Oord et al., 2016a*; *Van den Oord et al., 2016b*; *Kumar et al., 2019*), rather than modeling the scene in each video frame as a statistically independent event. Together these advances could lead to larger, higher-resolution, more reliable behavioral datasets that could revolutionize our understanding of relationships between behavior, the brain, and the environment.

In conclusion, we have presented a new toolkit, called DeepPoseKit, for automatically measuring animal posture from images. We combined recent advances from the literature to create methods that are fast, robust, and widely applicable to a range of species and experimental conditions. When designing our framework we emphasized usability across the entire software interface, which we expect will help to make these advanced tools accessible to a wider range of researchers. The fast inference and real-time capabilities of our methods should also help further reduce barriers to previously intractable questions across many scientific disciplines—including neuroscience, ethology, and behavioral ecology—both in the laboratory and the field.

## Materials and methods

We ran three main experiments to test and optimize our approach. First, we compared our new sub-pixel maxima layer to an integer-based global maxima with downsampled outputs ranging from $1\times$ to $\frac{1}{16}\times$ the input resolution using our Stacked DenseNet model. Next, we tested if training our

Stacked DenseNet model to predict the multi-scale geometry of the posture graph improves accuracy. Finally, we compared our model implementations of Stacked Hourglass and Stacked DenseNet to the models from *Pereira et al. (2019)* (LEAP) and *Mathis et al. (2018)* (DeepLabCut), which we also implemented in our framework (see Appendix 8 for details). We assessed both the inference speed and prediction accuracy of each model as well as training time and generalization ability. When comparing these models we incorporated the relevant improvements from our experiments—including subpixel maxima and predicting multi-scale geometry between keypoints—unless otherwise noted (see Appendix 8).

While we do make comparisons to the DeepLabCut model (*Mathis et al., 2018*) we do not use the same training routine as *Mathis et al. (2018)* and *Nath et al. (2019)*, who use binary crossentropy loss for optimizing the confidence maps in addition to the location refinement maps described by *Insafutdinov et al. (2016)*. We made this modification in order to hold the training routine constant for each model while only varying the model itself. However, we find that these differences between training routines effectively have no impact on performance when the models are trained using the same dataset and data augmentations (*Appendix 8—figure 1*). We also provide qualitative comparisons to demonstrate that, when trained with our DeepPoseKit framework, our implementation of the DeepLabCut model (*Mathis et al., 2018*) appears to produce fewer prediction errors than the original implementation from *Mathis et al. (2018)* and *Nath et al. (2019)* when applied to a novel video (*Appendix 8—figure 1—figure supplements 1* and *2*; *Appendix 8—figure 1—video 1*).

## Datasets

We performed experiments using the vinegar or 'fruit' fly (*Drosophila melanogaster*) dataset (*Figure 4*, *Figure 4—video 1*) provided by *Pereira et al. (2019)*, and to demonstrate the versatility of our methods we also compared model performance across two previously unpublished posture data sets from groups of desert locusts (*Schistocerca gregaria*) recorded in a laboratory setting (*Figure 4*, *Figure 4—video 2*), and herds of Grévy's zebras (*Equus grevyi*) recorded in the wild (*Figure 4*, *Figure 4—video 3*). The locust and zebra datasets are particularly challenging for pose estimation as they feature multiple interacting individuals—with focal individuals centered in the frame—and the latter with highly-variable environments and lighting conditions. These datasets are freely-available from https://github.com/jgraving/deepposekit-data (*Graving et al., 2019*; copy archived at https://github.com/elifesciences-publications/DeepPoseKit-Data).

Our locust dataset consisted of a group of 100 locusts in a circular plastic arena 1 m in diameter. The locust group was recorded from above using a high-resolution camera (Basler ace acA2040-90umNIR) and video recording system (Motif, loopbio GmbH). Locusts were localized and tracked using 2-D barcode markers (*Graving, 2017*) attached to the thorax with cyanoacrylate glue, and any missing localizations (<0.02% of the total dataset) between successful barcode reads were interpolated with linear interpolation. Our zebra dataset consisted of variably sized groups in the wild recorded from above using a commercially available quadcopter drone (DJI Phantom 4 Pro). Individual zebra were localized using custom deep-learning software based on Faster R-CNN (*Ren et al., 2015*) for predicting bounding boxes. The positions of each zebra were then tracked across frames using a linear assignment algorithm (*Munkres, 1957*) and data were manually verified for accuracy.

After positional tracking, the videos were then cropped using the egocentric coordinates of each individual and saved as separate videos—one for each individual. The images used for each training set were randomly selected using the k-means sampling procedure (with k = 10) described by *Pereira et al. (2019)* (Appendix 3) to reduce correlation between sampled images. After annotating the images with keypoints, we rotationally and translationally aligned the images and keypoints using the central body axis of the animal in each labeled image. This step allowed us to more easily perform data augmentations (see 'Model training') that allow the model to make accurate predictions regardless of the animal's body size and orientation (see Appendix 6). However, this preprocessing step is not a strict requirement for training, and there is no requirement for this preprocessing step when making predictions on new unlabeled data, such as with the methods described by *Pereira et al. (2019)* (Appendix 6). Before training each model we split each annotated dataset into randomly selected training and validation sets with 90% training examples and 10% validation examples, unless otherwise noted. The details for each dataset are described in *Table 1*.

**Table 1.** Datasets used for model comparisons.

| Name | Species | Resolution | # Images | # Keypoints | Individuals | Source |
|------|---------|-----------|----------|-------------|-------------|--------|
| Vinegar fly | *Drosophila melanogaster* | 192 × 192 | 1500 | **32** | Single | *Pereira et al., 2019* |
| Desert locust | *Schistocerca gregaria* | 160 × 160 | 800 | 35 | Multiple | This paper |
| Grévy's zebra | *Equus grevyi* | 160 × 160 | 900 | 9 | Multiple | This paper |

## Model training

For each experiment, we set our model hyperparameters to the same configuration for our Stacked DenseNet and Stacked Hourglass models. Both models were trained with $\frac{1}{4} \times$ resolution outputs and a stack of two networks with two outputs where loss was applied (see *Figure 2*). Although our model hyperparameters could be infinitely adjusted to trade off between speed and accuracy, we compared only one configuration for each of our model implementations. These results are not meant to be an exhaustive search of model configurations as the best configuration will depend on the application. The details of the hyperparameters we used for each model are described in Appendix 8.

To make our posture estimation tasks closer to realistic conditions, incorporate prior information (Appendix 3), and properly demonstrate the robustness of our methods to rotation, translation, scale, and noise, we applied various augmentations to each data set during training (*Figure 2*). All models were trained using data augmentations that included random flipping, or mirroring, along both the horizontal and vertical image axes with each axis being independently flipped by drawing from a Bernoulli distribution (with $p = 0.5$); random rotations around the center of the image drawn from a uniform distribution in the range $[-180°, +180°]$; random scaling drawn from a uniform distribution in the range [90%, 110%] for flies and locusts and [75%, 125%] for zebras (to account for greater size variation in the data set); and random translations along the horizontal and vertical axis independently drawn from a uniform distribution with the range $[-5\%, +5\%]$—where percentages are relative to the original image size. After performing these spatial augmentations we also applied a variety of noise augmentations that included additive noise (i.e., adding or subtracting randomly-selected values to pixels); dropout (i.e., setting individual pixels or groups of pixels to a randomly-selected value); blurring or sharpening (i.e., changing the composition of spatial frequencies); and contrast ratio augmentations—(i.e., changing the ratio between the highest pixel value and lowest pixel value in the image). These augmentations help to further ensure robustness to shifts in lighting, noise, and occlusions. See Appendix 3 for further discussion on data augmentation.

We trained our models (*Figure 2*) using mean squared error loss optimized using the ADAM optimizer (*Kingma and Ba, 2014*) with a learning rate of $1 \times 10^{-3}$ and a batch size of 16. We lowered the learning rate by a factor of five each time the validation loss did not improve by more than $1 \times 10^{-3}$ for 10 epochs. We considered models to be converged when the validation loss stopped improving for 50 epochs, and we calculated validation error as the Euclidean distance between predicted and ground-truth image coordinates for only the best performing version of the model, which we evaluated at the end of each epoch during optimization. We performed this procedure five times for each experiment and randomly selected a new training and validation set for each replicate.

## Model evaluation

Machine learning models are typically evaluated for their ability to generalize to new data, known as *predictive performance*, using a held-out *test set*—a subsample of annotated data that is not used for training or validation. However, due to the small size of the datasets used for making comparisons, we elected to use only a validation set for model evaluation, as using an overly small training or test set can bias assessments of a model's predictive performance (*Kuhn and Johnson, 2013*). Generally a test set is used to avoid biased performance measures caused by overfitting the model hyperparameters to the validation set. However, we did not adjust our model architecture to achieve better performance on our validation set—only to achieve fast inference speeds. While we did use validation error to decide when to lower the learning rate during training and when to stop training, lowering the learning rate in this way should have no effect on the generalization ability of the model, and because we heavily augment our training set during optimization—forcing the model to learn a much larger data distribution than what is included in the training and validation sets—

overfitting to the validation set is unlikely. We also demonstrate the generality of our results for each experiment by randomly selecting a new validation set with each replicate. All these factors make the Euclidean error for the unaugmented validation set a reasonable measure of the predictive performance for each model.

The inference speed for each model was assessed by running predictions on 100,000 randomly generated images with a batch size of 1 for real-time speeds and a batch size of 100 for offline speeds, unless otherwise noted. Our hardware consisted of a Dell Precision Tower 7910 workstation (Dell, Inc) running Ubuntu Linux v18.04 with 2× Intel Xeon E5-2623 v3 CPUs (8 cores, 16 threads at 3.00 GHz), 64 GB of RAM, a Quadro P6000 GPU and a Titan Xp GPU (NVIDIA Corporation). We used both GPUs (separately) for training models and evaluating predictive performance, but we only used the faster Titan Xp GPU for benchmarking inference speeds and training time. While the hardware we used for development and testing is on the high-end of the current performance spectrum, there is no requirement for this level of performance, and our software can easily be run on lower-end hardware. We evaluated inference speeds on multiple consumer-grade desktop computers and found similar performance (±10%) when using the same GPU; however, training speed depends more heavily other hardware components like the CPU and hard disk.

## Assessing prediction accuracy with Bayesian inference

To more rigorously assess performance differences between models, we parameterized the Euclidean error distribution for each experiment by fitting a Bayesian linear model with a Gamma-distributed likelihood function. This model takes the form:

$$p(y \mid X, \theta_\mu, \theta_\phi) \sim Gamma(\alpha, \beta)$$
$$\alpha = \mu^2 \phi^{-1}$$
$$\beta = \mu \phi^{-1}$$
$$\mu = h(X\theta_\mu)$$
$$\phi = h(X\theta_\phi)$$

where $X$ is the design matrix composed of binary indicator variables for each pose estimation model, $\theta_\mu$ and $\theta_\phi$ are vectors of intercepts, $h(\cdot)$ is the softplus function (*Dugas et al., 2001*)—or $h(x) = \log(1 + e^x)$—used to enforce positivity of $\mu$ and $\phi$, and $y$ is the Euclidean error of the pose estimation model. Parameterizing our error distributions in this way allows us to calculate the posterior distributions for the mean $\mathrm{E}[y] = \alpha\beta^{-1} \equiv \mu$ and variance $\mathrm{Var}[y] = \alpha\beta^{-2} \equiv \phi$. This parameterization then provides us with a statistically rigorous way to assess differences in model accuracy in terms of both central tendency and spread—accounting for both epistemic uncertainty (unknown unknowns; e.g., parameter uncertainty) and aleatoric uncertainty (known unknowns; e.g., data variance). Details of how we fitted these models can be found in Appendix 7.

## Acknowledgements

We are indebted to Talmo Pereira et al. and A Mathis et al. for making their software open-source and freely-available—this project would not have been possible without them. We also thank M Mathis and A Mathis for their comments, which greatly improved the manuscript. We thank François Chollet, the Keras and TensorFlow teams, and Alexander Jung for their open source contributions, which provided the core programming interface for our work. We thank A Strandburg-Peshkin, Vivek H Sridhar, Michael L Smith, and Joseph B Bak-Coleman for their helpful discussions on the project and comments on the manuscript. We also thank MLS for the use of his GPU. We thank Felicitas Oehler for annotating the zebra posture data and Chiara Hirschkorn for assistance with filming the locusts and annotating the locust posture data. We thank Alex Bruttel, Christine Bauer, Jayme Weglarski, Dominique Leo, Markus Miller and loobio GmbH for providing technical support. We acknowledge the NVIDIA Corporation for their generous donations to our research. This project received funding from the European Union's Horizon 2020 research and innovation programme under the Marie Sklodowska-Curie grant agreement No. 748549. BRC acknowledges support from the University of Konstanz Zukunftskolleg's Investment Grant program. IDC acknowledges support from NSF Grant IOS-1355061, Office of Naval Research Grants N00014-09-1-1074 and N00014-14-

1-0635, Army Research Office Grants W911NG-11-1-0385 and W911NF14-1-0431, the Struktur-und Innovationsfonds fur die Forschung of the State of Baden-Württemberg, the DFG Centre of Excellence 2117 'Centre for the Advanced Study of Collective Behaviour' (ID: 422037984), and the Max Planck Society.

## Additional information

### Competing interests

Iain D Couzin: Reviewing editor, *eLife*. The other authors declare that no competing interests exist.

### Funding

| Funder | Grant reference number | Author |
| --- | --- | --- |
| National Science Foundation | IOS-1355061 | Iain D Couzin |
| Office of Naval Research | N00014-09-1-1074 | Iain D Couzin |
| Office of Naval Research | N00014-14-1-0635 | Iain D Couzin |
| Army Research Office | W911NG-11-1-0385 | Iain D Couzin |
| Army Research Office | W911NF14-1-0431 | Iain D Couzin |
| Deutsche Forschungsgemeinschaft | DFG Centre of Excellence 2117 | Iain D Couzin |
| University of Konstanz | Zukunftskolleg Investment Grant | Blair R Costelloe |
| Ministry of Science, Research and Art Baden-Württemberg | The Strukture-und Innovations fonds fur die Forschung of the State of Baden-Wurttemberg | Iain D Couzin |
| Max Planck Society | | Iain D Couzin |
| Horizon 2020 Framework Programme | Marie Sklodowska-Curie grant agreement No. 748549 | Blair R Costelloe |
| Nvidia | GPU Grant | Blair R Costelloe |
| Nvidia | GPU Grant | Liang Li |

The funders had no role in study design, data collection and interpretation, or the decision to submit the work for publication.

### Author contributions

Jacob M Graving, Conceptualization, Data curation, Software, Formal analysis, Validation, Investigation, Visualization, Methodology, Writing—original draft, Project administration, Writing—review and editing; Daniel Chae, Software, Formal analysis, Methodology, Writing—original draft; Hemal Naik, Conceptualization, Methodology, Writing—review and editing; Liang Li, Data curation, Validation, Writing—review and editing; Benjamin Koger, Data curation, Investigation, Writing—review and editing; Blair R Costelloe, Data curation, Supervision, Project administration, Writing—review and editing; Iain D Couzin, Conceptualization, Resources, Supervision, Funding acquisition, Project administration, Writing—review and editing

### Author ORCIDs

Jacob M Graving  https://orcid.org/0000-0002-5826-467X
Liang Li  https://orcid.org/0000-0002-2447-6295
Blair R Costelloe  https://orcid.org/0000-0001-5291-788X
Iain D Couzin  https://orcid.org/0000-0001-8556-4558

## Ethics

Animal experimentation: All procedures for collecting the zebra (*E. grevyi*) dataset were reviewed and approved by Ethikrat, the independent Ethics Council of the Max Planck Society. The zebra dataset was collected with the permission of Kenya's National Commission for Science, Technology and Innovation (NACOSTI/P/17/59088/15489 and NACOSTI/P/18/59088/21567) using drones operated by BRC with the permission of the Kenya Civil Aviation Authority (authorization numbers: KCAA/OPS/2117/4 Vol. 2 (80), KCAA/OPS/2117/4 Vol. 2 (81), KCAA/OPS/2117/5 (86) and KCAA/OPS/2117/5 (87); RPAS Operator Certificate numbers: RPA/TP/0005 AND RPA/TP/000-0009).

## Decision letter and Author response

Decision letter https://doi.org/10.7554/eLife.47994.sa1
Author response https://doi.org/10.7554/eLife.47994.sa2

# Additional files

## Supplementary files

• Transparent reporting form

## Data availability

Data used and generated for experiments and model comparisons are included in the supporting files. Posture datasets can be found at https://github.com/jgraving/deepposekit-data (copy archived at https://github.com/elifesciences-publications/DeepPoseKit-Data). The code for DeepPoseKit is publicly available at the URL we provided in the paper: https://github.com/jgraving/deepposekit/ (copy archived at https://github.com/elifesciences-publications/DeepPoseKit).

The following dataset was generated:

| Author(s) | Year | Dataset title | Dataset URL | Database and Identifier |
| --- | --- | --- | --- | --- |
| Graving JM, Chae D, Naik H, Li L, Koger B, Costelloe BR, Couzin IA | 2019 | Example Datasets for DeepPoseKit (Version v0.1-doi) [Data set]. | http://doi.org/10.5281/zenodo.3366908 | Zenodo, 10.5281/zenodo.3366908 |

The following previously published dataset was used:

| Author(s) | Year | Dataset title | Dataset URL | Database and Identifier |
| --- | --- | --- | --- | --- |
| Pereira TD, Aldarondo DE, Willmore L, Kislin M, Wang SS-H, Murthy M, Shaevitz JW | 2018 | Fast animal pose estimation using deep neural networks | https://dataspace.princeton.edu/jspui/handle/88435/dsp01pz50gz79z | DataSpace, dsp01pz50gz79z |

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

# Appendix 1

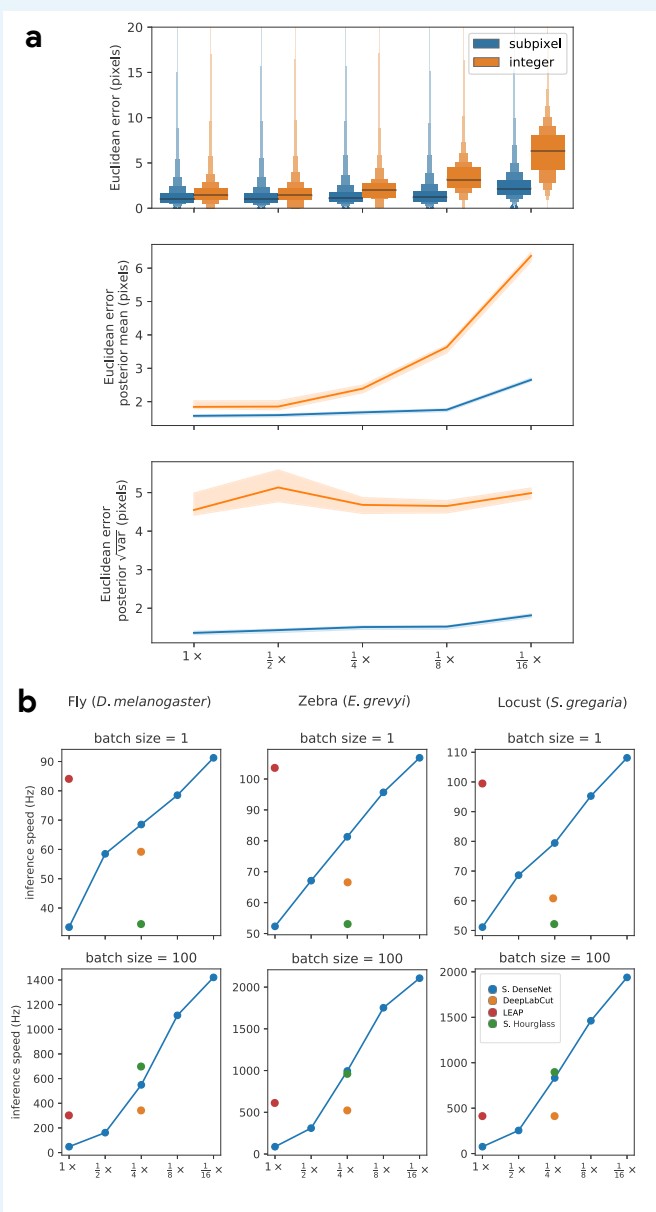

**Appendix 1—figure 1.** Our subpixel maxima algorithm increases speed without decreasing accuracy. Prediction accuracy on the fly dataset is maintained across downsampling configurations (**a**). Letter-value plots (a-top) show the raw error distributions for each configuration. Visualizations of the credible intervals (99% highest-density region) of the posterior distributions for the mean and variance (a-bottom) illustrate statistical differences between the error distributions, where using subpixel maxima decreases both the mean and variance of the error distribution. Inference speed is fast and can be run in real-time on single images (batch size = 1) at ~30–110 Hz or offline (batch size = 100) upwards of 1000 Hz (**b**). Plots show the inference speeds for our Stacked DenseNet model across downsampling configurations as well as the other models we tested for each of our datasets.

The online version of this article includes the following source data is available for figure 1:

**Appendix 1—figure 1—source data 1.** Raw prediction errors for experiments in *Appendix 1—figure 1a*.

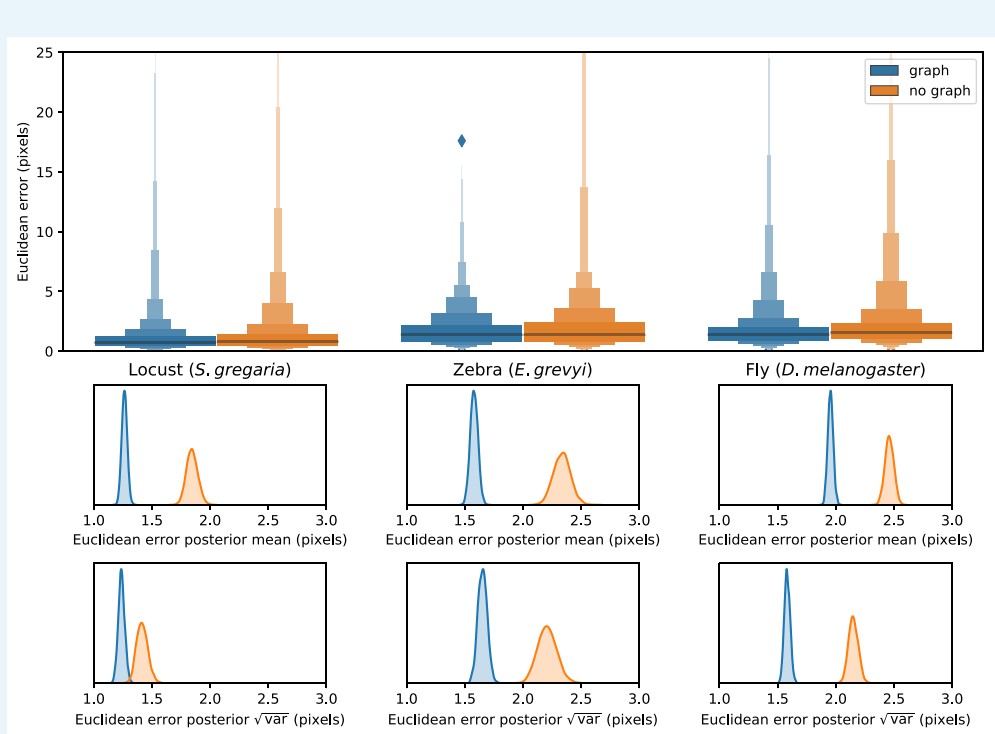

**Appendix 1—figure 2.** Predicting the multi-scale geometry of the posture graph reduces error. Letter-value plots (top) show the raw error distributions for each experiment. Visualizations of the posterior distributions for the mean and variance (bottom) show statistical differences between the error distributions. Predicting the posture graph decreases both the mean and variance of the error distribution.

The online version of this article includes the following source data is available for figure 2:

**Appendix 1—figure 2—source data 1.** Raw prediction errors for experiments in *Appendix 1—figure 2*.

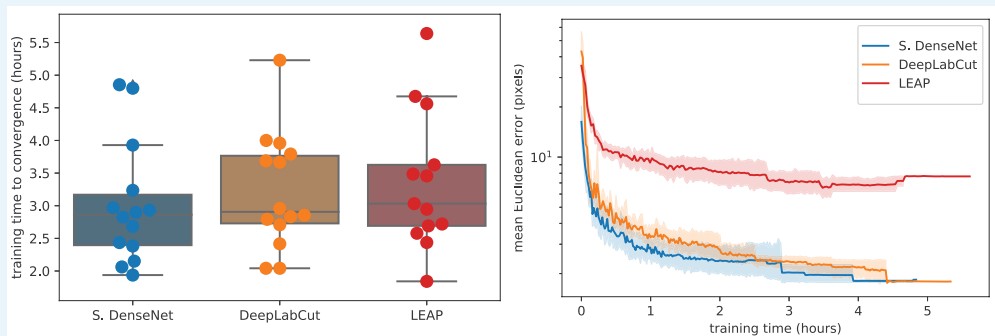

**Appendix 1—figure 3.** Training time required for our Stacked DenseNet model, the DeepLab-Cut model (*Mathis et al., 2018*), and the LEAP model (*Pereira et al., 2019*) (n = 15 per model) using our zebra dataset. Boxplots and swarm plots (left) show the total training time to convergence (<0.001 improvement in validation loss for 50 epochs). Line plots (right) illustrate the Euclidean error of the validation set during training, where error bars show bootstrapped (n = 1000) 99% confidence intervals of the mean. Fully training models to convergence requires only a few hours of optimization (left) with reasonable accuracy reached after only 1 hr (right) for our Stacked DenseNet model.

The online version of this article includes the following source data is available for figure 3:

**Appendix 1—figure 3—source data 1.** Total training time for each model in *Appendix 1—*

*figure 3*.
**Appendix 1—figure 3—source data 2.** Mean euclidean error as a function of training time for each model in *Appendix 1—figure 3*.

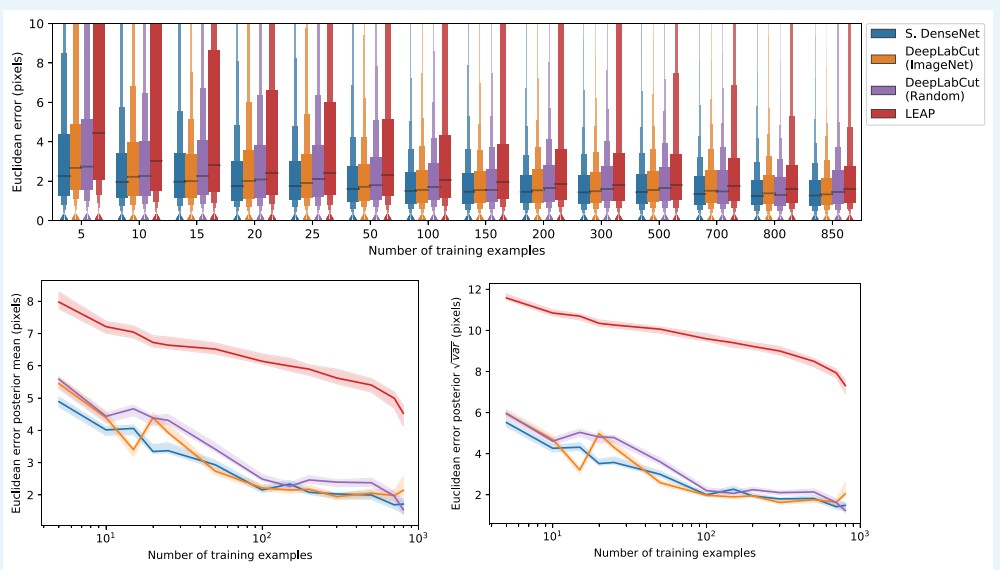

**Appendix 1—figure 4.** A comparison of prediction accuracy with different numbers of training examples from our zebra dataset. The error distributions shown as letter-value plots (top) illustrate the Euclidean error for the remainder of the dataset not used for training—with a total of 900 labeled examples in the dataset. Line plots (bottom) show posterior credible intervals (99% highest-density region) for the mean and variance of the error distributions. We tested our Stacked DenseNet model; the DeepLabCut model (*Mathis et al., 2018*) with transfer learning—that is with weights pretrained on ImageNet (*Deng et al., 2009*); the same model without transfer learning—that is with randomly-initialized weights; and the LEAP model (*Pereira et al., 2019*). Our Stacked DenseNet model achieves high accuracy using few training examples without the use the transfer learning. Using pretrained weights does slightly decrease overall prediction error for the DeepLabCut model (*Mathis et al., 2018*), but the effect size is relatively small.
The online version of this article includes the following source data is available for figure 4:
**Appendix 1—figure 4—source data 1.** Raw prediction errors for experiments in *Appendix 1—figure 4*.

## Appendix 2

# Convolutional neural networks (CNNs)

*Artificial neural networks* like CNNs are complex, non-linear regression models that 'learn' a hierarchically-organized set of parameters from real-world data via optimization. These machine learning models are now commonplace in science and industry and have proven to be surprisingly effective for a large number of applications where more conventional statistical models have failed (**LeCun et al., 2015**). For computer vision tasks, CNN parameters typically take the form of two-dimensional convolutional filters that are optimized to detect spatial features needed to model relationships between high-dimensional image data and some related variable(s) of interest, such as locations in space—for example posture keypoints—or semantic labels (**Long et al., 2015**; **Badrinarayanan et al., 2015**).

Once a training set is generated (Appendix 3), a CNN model must be selected and optimized to perform the prediction task. CNNs are incredibly flexible with regard to how models are specified and trained, which is both an advantage and a disadvantage. This flexibility means models can be adapted to almost any computer vision task, but it also means the number of possible model architectures and optimization schemes is very large. This can make selecting an architecture and specifying hyperparameters a challenging process. However, most research on pose estimation has converged on a set of models that generally work well for this task (Appendix 4).

After selecting an architecture, the parameters of the model are set to an initial value and then iteratively updated to minimize some objective function, or *loss function*, that describes the difference between the model's predictive distribution and the true distribution of the data—in other words, the likelihood of the model's output is maximized. These parameter updates are performed using a modified version of the gradient descent algorithm (**Cauchy, 1847**) known as *mini-batch stochastic gradient descent*—often referred to as simply *stochastic gradient descent* or *SGD* (**Robbins and Monro, 1951**; **Kiefer and Wolfowitz, 1952**). SGD iteratively optimizes the model parameters using small randomly-selected subsamples, or *batches*, of training data. Using SGD allows the model to be trained on extremely large datasets in an iterative 'online' fashion without the need to load the entire dataset into memory. The model parameters are updated with each batch by adjusting the parameter values in a direction that minimizes the error—where one round of training on the full dataset is commonly referred to as an *epoch*. The original SGD algorithm requires careful selection and tuning of hyperparameters to successfully optimize a model, but modern versions of the algorithm, such as *ADAM* (**Kingma and Ba, 2014**), automatically tune these hyperparameters, which makes optimization more straightforward.

The model parameters are optimized until they reach a convergence criterion, which is some measure of performance that indicates the model has reached a good location in parameter space. The most commonly used convergence criterion is a measure of predictive accuracy—often the loss function used for optimization—on a held-out *validation set*—a subsample of the training data not used for optimization—that evaluates the model's ability to generalize to new 'out-of-sample' data. The model is typically evaluated at the end of each training epoch to assess performance on the validation set. Once performance on the validation set stops improving, training is usually stopped to prevent the model from overfitting to the training set—a technique known as *early stopping* (**Prechelt, 1998**).

## Collecting training data

Depending on the variability of the data, CNNs usually require thousands or tens of thousands of manually-annotated examples in order to reach human-level accuracy. However, in laboratory settings, sources of image variation like lighting and spatial scale can be more easily controlled, which minimizes the number of training examples needed to achieve accurate predictions.

This need for a large training set can be further reduced in a number of ways. Two commonly used methods include (1) *transfer learning*—using a model with parameters that are pre-trained on a larger set of images, such as the ImageNet database (*Deng et al., 2009*), containing diverse features (*Pratt, 1992*; *Insafutdinov et al., 2016*; *Mathis et al., 2018*)— and (2) *augmentation*— artificially increasing data variance by applying spatial and noise transformations such as flipping (mirroring), rotating, scaling, and adding different forms of noise or artificial occlusions. Both of these methods act as useful forms of *regularization*— incorporating a prior distribution—that allows the model to generalize well to new data even when the training set is small. Transfer learning incorporates prior information that images from the full dataset should contain statistical features similar to other images of the natural world, while augmentation incorporates prior knowledge that animals are bilaterally symmetric, can vary in their body size, position, and orientation, and that noise and occlusions sometimes occur.

*Pereira et al. (2019)* introduced two especially clever solutions for collecting an adequate training set. First, they cluster unannotated images based on pixel variance and uniformly sample images from each cluster, which reduces correlation between training examples and ensures the training data are representative of the entire distribution of possible images. Second, they use *active learning* where a CNN is trained on a small number of annotated examples and is then used to initialize keypoint locations for a larger set of unannotated data. These pre-initialized data are then manually corrected by the annotator, the model is retrained, and the unannotated data are re-initialized. The annotator applies this process iteratively as the training set grows larger until they are providing only minor adjustments to the pre-initialized data. This 'human-in-the-loop'-style annotation expedites the process of generating an adequately large training set by reducing the cognitive load on the annotator— where the pose estimation model serves as a 'cognitive partner'. Such a strategy also allows the annotator to automatically select new training examples based on the performance of the current iteration—where low-confidence predictions indicate examples that should be annotated for maximum improvement (*Figure 1*).

Of course, annotating image data requires software made for this purpose. *Pereira et al. (2019)* provide a custom annotation GUI written in MATLAB specifically designed for annotating posture using an active learning strategy. recently *Mathis et al. (2018)* added a Python-based GUI in an updated version of their software—including active learning and image sampling methods (see *Nath et al., 2019*). Our framework also includes a Python-based GUI for annotating data with similar features.

## Appendix 4

## Fully-convolutional regression

For the task of pose estimation, a CNN is optimized to predict the locations of postural keypoints in an image. One approach is to use a CNN to directly predict the numerical value of each keypoint coordinate as an output. However, making predictions in this way removes real-world constraints on the model's predictive distribution by destroying spatial relationships within images, which negates many of the advantages of using CNNs in the first place.

CNNs are particularly good at transforming one image to produce another related image, or set of images, while preserving spatial relationships and allowing for translation-invariant predictions—a configuration known as a *fully-convolutional neural network* or *F-CNN* (*Long et al., 2015*). Therefore, instead of directly regressing images to coordinate values, a popular solution (*Newell et al., 2016*; *Insafutdinov et al., 2016*; *Mathis et al., 2018*; *Pereira et al., 2019*) is to optimize a F-CNN that transforms images to predict a stack of output images known as *confidence maps*—one for each keypoint. Each confidence map in the output volume contains a single, two-dimensional, symmetric Gaussian indicating the location of each joint, and the scalar value of the peak indicates the confidence score of the prediction—typically a value between 0 and 1. The confidence maps are then processed to produce the coordinates of each keypoint.

In the case of *multiple pose estimation* where an image contains many individuals, the global geometry of the posture graph is also predicted by training the model to produce *part affinity fields* (*Cao et al., 2017*)— directional vector fields drawn between joints in the posture graph—or *pairwise terms* (*Insafutdinov et al., 2016*)—vector fields of the conditional distributions between posture keypoints (e.g., $p(foot|head)$). This allows multiple posture graphs to be disentangled from the image using graph partitioning as the vector fields indicate the probability of the connection between joints (see *Cao et al., 2017* for details).

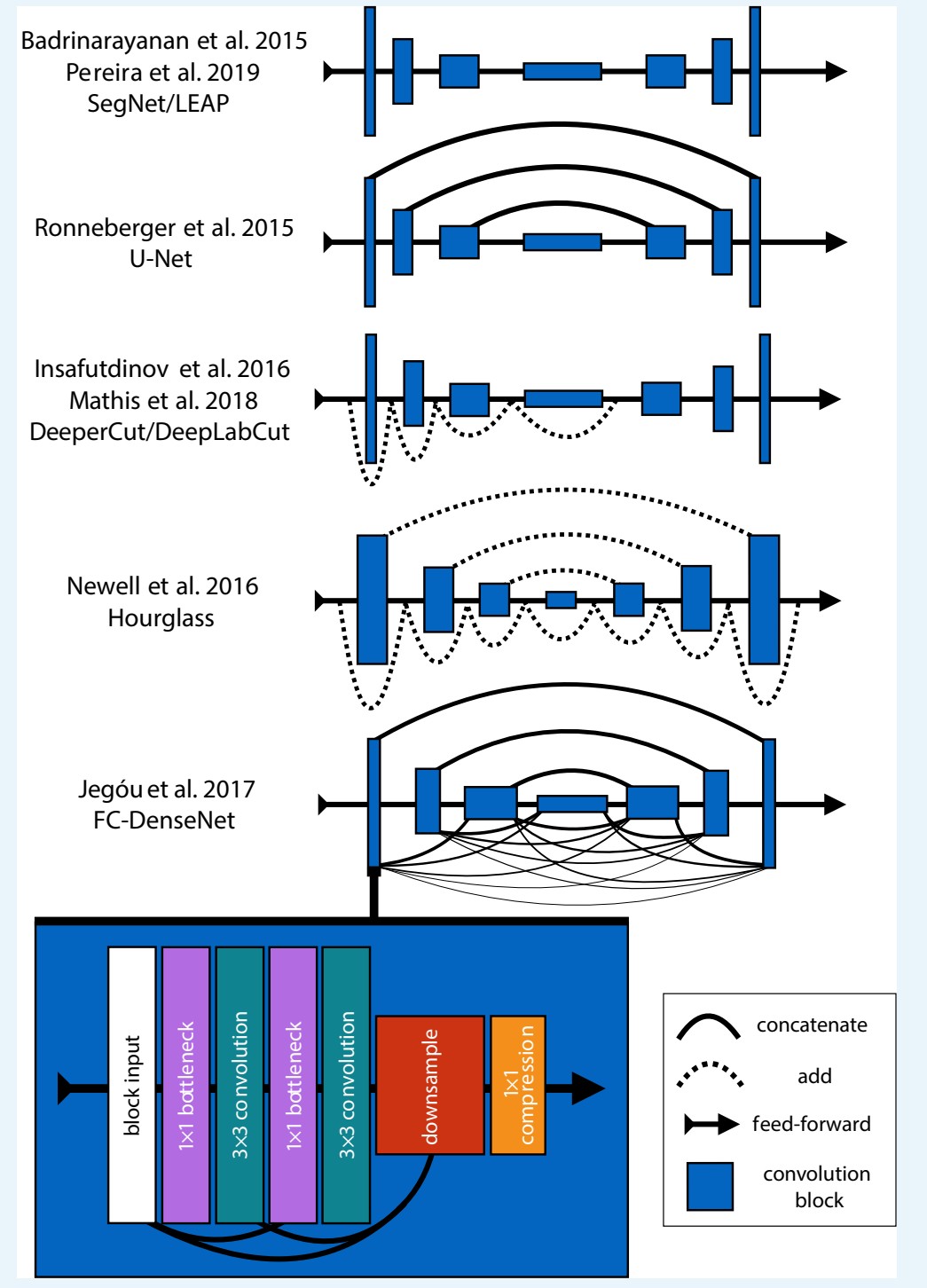

**Appendix 4—figure 1.** An illustration showing the progression of encoder-decoder architectures from the literature—ordered by performance from top to bottom (see Appendix 4: 'Encoder-decoder models' for further details). Most advances in performance have come from adding connections between layers in the network, culminating in FC-DenseNet from *Jégou et al. (2017)*. Lines in each illustration indicate connections between convolutional blocks with the thickness of the line indicating the magnitude of information flow between layers in the network. The size of each convolution block indicates the relative number of feature maps (width) and spatial scale (height). The callout for FC-DenseNet (*Jégou et al., 2017*; bottom-left) shows the elaborate set of skip connections within each densely-connected

convolutinal block as well as our additions of bottleneck and compression layers (described by *Huang et al., 2017a*) to increase efficiency (Appendix 8).

## Encoder-decoder models

A popular type of F-CNN (Appendix 4) for solving posture regression problems is known as an *encoder-decoder* model (*Appendix 4—figure 2*), which first gained popularity for the task of *semantic segmentation*—a supervised computer vision problem where each pixel in an image is classified into a one of several labeled categories like 'dog', 'tree', or 'road' (*Long et al., 2015*). This model is designed to repeatedly convolve and downsample input images in the bottom-up *encoder* step and then convolve and upsample the encoder's output in the top-down *decoder* step to produce the final output. Repeatedly applying convolutions and non-linear functions, or *activations*, to the input images transforms pixel values into higher-order spatial features, while downsampling and upsampling respectively increases and decreases the scale and complexity of these features.

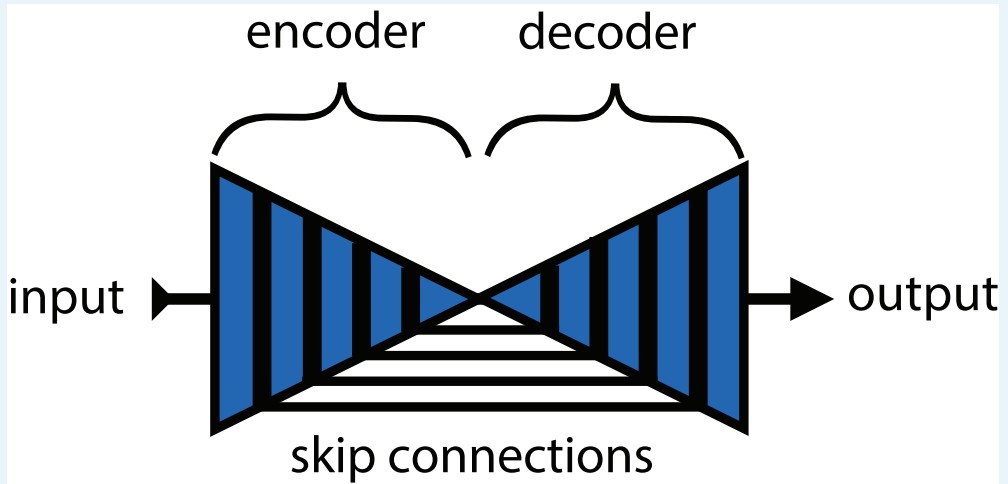

**Appendix 4—figure 2.** An illustration of the basic encoder-decoder design. The encoder converts the input images into spatial features, and the decoder transforms spatial features to the desired output.

*Badrinarayanan et al. (2015)* were the first to popularize a form of this model —known as *SegNet*— for semantic segmentation. However, this basic design is inherently limited because the decoder relies solely on the downsampled output from the encoder, which restricts the features used for predictions to those with the largest spatial scale and highest complexity. For example, a very deep network might learn a complex spatial pattern for predicting 'grass' or 'trees', but because it cannot directly access information from the earliest layers of the network, it cannot use the simplest features that plants are green and brown. Subsequent work by *Ronneberger et al. (2015)* improved on these problems with the addition of *skip connections* between the encoder and decoder, where feature maps from encoder layers are concatenated to those decoder layers with the same spatial scale. This set of connections then allows the optimizer, rather than the user, to select the most relevant spatial scale(s) for making predictions.

*Jégou et al. (2017)* are the latest to advance the encoder-decoder paradigm. These researchers introduced a fully-convolutional version of *Huang et al. (2017a)* *DenseNet* architecture known as a *fully-convolutional DenseNet*, or *FC-DenseNet*. FC-DenseNet's key improvement is an elaborate set of feed-forward residual connections where the input to each convolutional layer is a concatenated stack of feature maps from all previous layers. This densely-connected design was motivated by the insight that many state-of-the-art models learn a large proportion of redundant features. Most CNNs are not designed so that the final output layers can access all feature maps in the network simultaneously, and this limitation

causes these networks to 'forget' and 'relearn' important features as the input images are transformed to produce the output. In the case of the incredibly popular ResNet-101 (*He et al., 2016*) nearly 40% of the features can be classified as redundant (*Ayinde and Zurada, 2018*). A densely-connected architecture has the advantages of reduced feature redundancy, increased feature reuse, enhanced feature propagation from early layers to later layers, and subsequently, a substantial reduction in the number of parameters needed to achieve state-of-the-art results (*Huang et al., 2017a*). Recent work has also shown that DenseNet's elaborate set of skip connections have the pleasant side-effect of convexifying the loss landscape during optimization (*Li et al., 2018*), which allows for faster optimization and increases the likelihood of reaching a good optimum.

## Appendix 5

### The state of the art for individual pose estimation

Many of the current state-of-the-art models for individual posture estimation are based on the design from **Newell et al. (2016)** (e.g., **Ke et al., 2018**; **Chen et al., 2017**; also see benchmark results from **Andriluka et al. (2014)**, but employ various modifications that increase complexity to improve performance. **Newell et al. (2016)** employ what they call a *Stacked Hourglass* network (**Appendix 4—figure 1**), which consists of a series of multi-scale encoder-decoder *hourglass* modules connected together in a feed-forward configuration (**Figure 2**). The main novelties these researchers introduce include (1) stacking multiple hourglass networks together for repeated top-down-bottom-up inference, (2) using convolutional blocks based on the ResNet architecture (**He et al., 2016**) with residual connections between the input and output of each block, and (3) using residual connections between the encoder and decoder (similar to **Ronneberger et al., 2015**) with residual blocks in between. **Newell et al. (2016)** also apply a technique known as *intermediate supervision* (**Figure 2**) where the loss function used for model training is applied to the output of each hourglass as a way of improving optimization across the model's many layers. Recent work by **Jégou et al. (2017)** has further improved on this encoder-decoder design (see Appendix 4: 'Encoder-decoder models' and **Appendix 4—figure 1**), but to the best of our knowledge, the model introduced by **Jégou et al. (2017)** has not been previously applied to pose estimation.

**Appendix 6**

## Overparameterization and the limitations of LEAP

Overparameterization is a key limitation for many pose estimation methods, and addressing this problem is critical for high-performance applications. *Pereira et al. (2019)* approached this problem by designing their LEAP model after the model from *Badrinarayanan et al. (2015)*, which is a straighforward encoder-decoder design (*Appendix 4—figure 1*; Appendix 4: 'Encoder-decoder models'). They benchmarked their model on posture estimation tasks for laboratory animals and compared performance with the more-complex Stacked Hourglass model from *Newell et al. (2016)*. They found their smaller, simplified model achieved equal or better median accuracy with dramatic improvements in inference speed up to 185 Hz. However, *Pereira et al. (2019)* first rotationally and translationally aligned each image to improve performance, and their reported inference speeds do not include this computationally expensive preprocessing step. Additionally, rotationally and translationally aligning images is not always possible when the background is complex or highly-variable—such as in field settings—or the study animal has a non-rigid body. This limitation makes the LEAP model (*Pereira et al., 2019*) unsuitable in many cases. While their approach is simple and effective for a multitude of experimental setups, the LEAP model (*Pereira et al., 2019*) is also implicitly limited in the same ways as *Badrinarayanan et al. (2015)*'s SegNet model (see Appendix 4: 'Encoder-decoder models'). The LEAP model cannot make predictions using multiple spatial scales and is not robust to data variance such as rotations (*Pereira et al., 2019*).

## Appendix 7

### Linear model fitting with Stan

We estimated the joint posterior $p(\theta_\mu, \theta_\phi \mid X, y)$ for each model using the No-U-Turn Sampler (NUTS; *Hoffman and Gelman, 2014*), a self-tuning variant of the Hamiltonian Monte Carlo (HMC) algorithm (*Duane et al., 1987*), implemented in Stan (*Carpenter et al., 2017*). We drew HMC samples using four independent Markov chains consisting of 1000 warm-up iterations and 1000 sampling iterations for a total of 4000 sampling iterations. To speed up sampling, we randomly subsampled 20% of the data from each replicate when fitting each linear model, and we fit each model 5 times to ensure the results were consistent. All models converged without any signs of pathological behavior. We performed a posterior predictive check by visually inspecting predictive samples to assess model fit. For our priors we chose relatively uninformative distributions $\theta_\mu \sim Cauchy(0, 5)$ and $\theta_\phi \sim Cauchy(0, 10)$, but we found that the choice of prior generally did not have an effect on the final result due to the large amount of data used to fit each model.

**Appendix 8**

## Stacked DenseNet

Our Stacked DenseNet model consists of an initial $7 \times 7$ convolutional layer with stride 2, to efficiently downsample the input resolution—following *Newell et al. (2016)*—followed by a stack of densely-connected hourglass networks with intermediate supervision (Appendix 5) applied at the output of each network. We also include hyperparameters for the bottleneck and compression layers described by *Huang et al. (2017a)* to make the model as efficient as possible. These consist of applying a $1 \times 1$ convolution to inexpensively compress the number of feature maps before each $3 \times 3$ convolution as well as when downsampling and upsampling (see *Huang et al., 2017a* and *Appendix 4—figure 1* for details).

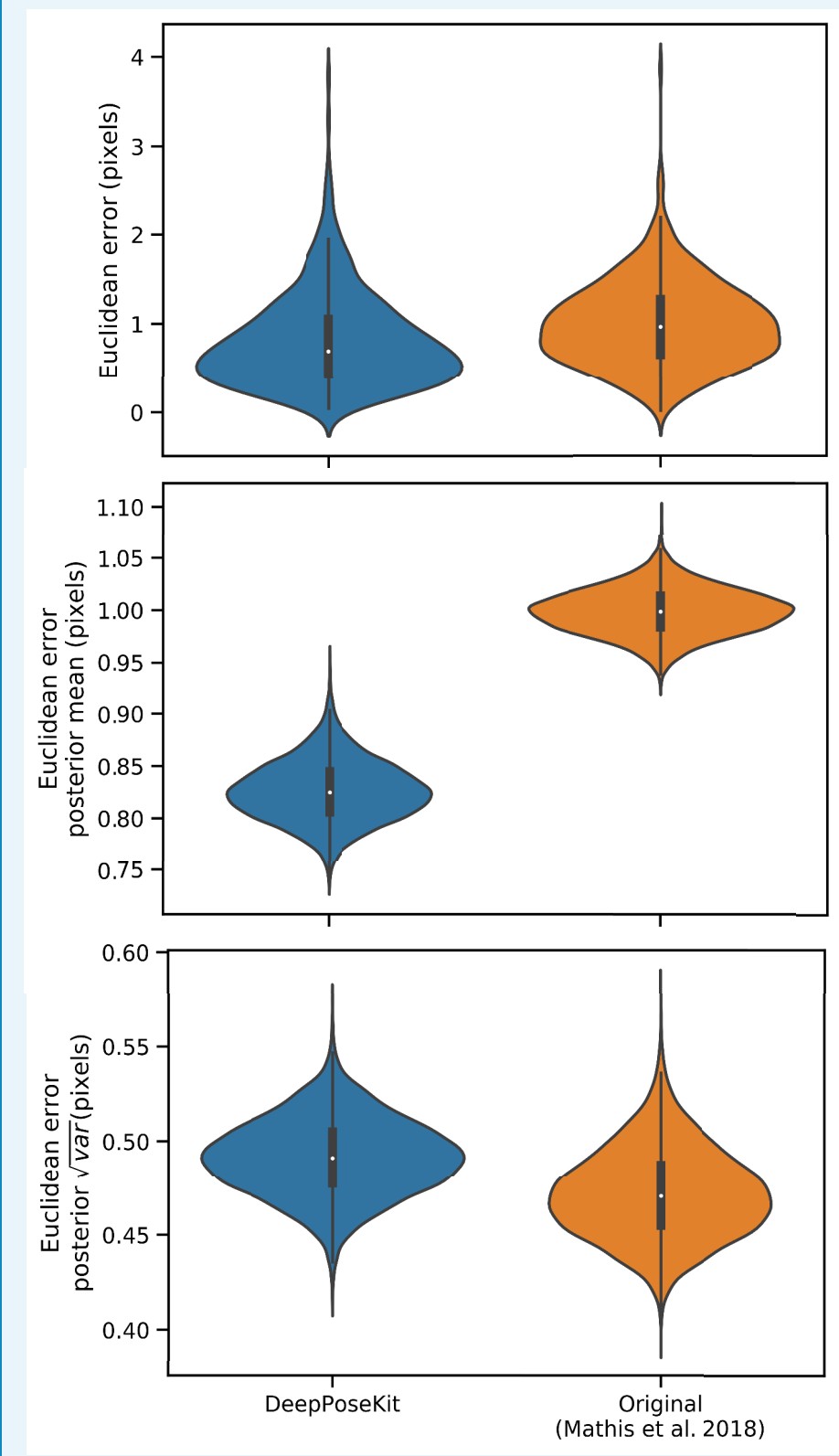

**Appendix 8—figure 1.** Prediction errors for the odor-trail mouse dataset from *Mathis et al.* *(2018)* using the original implementation of the DeepLabCut model (*Mathis et al., 2018*; *Nath et al., 2019*) and our modified version of this model implemented in DeepPoseKit. Mean prediction error is slightly lower for the DeepPoseKit implementation, but there is no

discernible difference in variance. These results indicate that the models achieve nearly identical prediction accuracy despite modification. We also provide qualitative comparisons of these results in *Appendix 8—figure 1—figure supplement 1* and *2*, and *Appendix 8—figure 1—video 1*.

The online version of this article includes the following video and source data for figure 1:

**Appendix 8—figure 1—source data 1.** Raw prediction errors for our DeepLabCut model (*Mathis et al., 2018*) reimplemented in DeepPoseKit in *Appendix 8—figure 1*.
**Appendix 8—figure 1—source data 2.** Raw prediction errors for the original DeepLabCut model from *Mathis et al. (2018)* in *Appendix 8—figure 1*.

**Appendix 8—figure 1—video 1.** A video comparison of the tracking output of our implementation of the DeepLabCut model (*Mathis et al., 2018*) in DeepPoseKit vs. the original implementation from *Mathis et al. (2018)* and *Nath et al. (2019)*.

**Appendix 8—Figure 1 supplement 1.** Plots of the predicted output for *Appendix 8—figure 1—video 1* comparing our implementation of the DeepLabCut model (*Mathis et al., 2018*) in DeepPoseKit *vs.* the original implementation from *Mathis et al. (2018)*, and *Nath et al. (2019)*.
**Appendix 8—Figure 1 supplement 2.** Plots of the temporal derivatives of the predicted output for *Appendix 8—figure 1—video 1* comparing our implementation of the DeepLabCut model (*Mathis et al., 2018*) in DeepPoseKit vs. the original implementation from *Mathis et al. (2018)*, and *Nath et al. (2019)*.

## Model hyperparameters

For our Stacked Hourglass model we used a block size of 64 filters (64 filters per $3 \times 3$ convolution) with a bottleneck factor of 2 ($64/2 = 32$ filters per $1 \times 1$ bottleneck block). For our Stacked DenseNet model we used a growth rate of 48 (48 filters per $3 \times 3$ convolution), a bottleneck factor of 1 ($1 \times$ growth rate = 48 filters per $1 \times 1$ bottleneck block), and a compression factor of 0.5 (feature maps compressed with $1 \times 1$ convolution to 0.5 m when upsampling and downsampling, where $m$ is the number of feature maps). For our Stacked DenseNet model we also replaced the typical configuration of batch normalization and ReLU activations (*Goodfellow et al., 2016*) with the more recently-developed self-normalizing SELU activation function (*Klambauer et al., 2017*), as we found this modification increased inference speed. For the LEAP model (*Pereira et al., 2019*) we used a $1 \times$ resolution output with integer-based global maxima because we wanted to compare our more complex models with this model in the original configuration described by *Pereira et al. (2019)*. The LEAP model could be modified to output smaller confidence maps and increase inference speed, but because there is no obvious 'best' way to alter the model to achieve this, we forgo any modification. Additionally, applying our subpixel maxima algorithm at high-resolution reduces inference speed compared to integer-based maxima, so this would bias our speed comparisons.

## Our implementation of the DeepLabCut model

Because the DeepLabCut model from *Mathis et al. (2018)* was not implemented in Keras (a requirement for our pose estimation framework), we re-implemented it. Implementing this model directly in our framework is important to ensure model training and data augmentation are identical when making comparisons between models. As a consequence, our version of this model does not exactly match the description in the paper but is identical except for the output. Rather than using the location refinement maps described by *Insafutdinov et al. (2016)* and post-processing confidence maps on the CPU, our version of the DeepLabCut model (*Mathis et al., 2018*) has an additional transposed convolutional layer to upsample the output to $\frac{1}{4} \times$ resolution and uses our subpixel maxima algorithm.

To demonstrate that our implementation of the DeepLabCut model matches the performance described by *Mathis et al. (2018)*, we compared prediction accuracy between the two frameworks using the odor-trail mouse dataset provided by *Mathis et al. (2018)* (downloaded from https://github.com/AlexEMG/DeepLabCut/). This dataset consists of 116 images of a freely moving individual mouse labeled with four keypoints describing the location of the snout, ears, and the base of the tail. See *Mathis et al. (2018)* for further details on this

dataset. We trained both models using 95% training and 5% validation data and applied data augmentations for both frameworks using the data augmentation procedure described by *Nath et al. (2019)*. We tried to match these data augmentations as best as possible in DeepPoseKit; however, rather than cropping images as described by *Nath et al. (2019)*, we randomly translated the images independently along the horizontal and vertical axis by drawing from a uniform distribution in the range [−100%, +100%]—where percentages are relative to the size of each axis. Translating the images in this way should serve the same purpose as cropping them.

We trained the original DeepLabCut model (*Mathis et al., 2018*) using the default settings and recommendations from *Nath et al. (2019)* for 1 million training iterations. See *Mathis et al. (2018)*; *Nath et al. (2019)* for further details on the data augmentation and training routine for the original implementation of the DeepLabCut model (*Mathis et al., 2018*). For our re-implementation of the DeepLabCut model (*Mathis et al., 2018*), we trained the model with the same batch size and optimization scheme described in the 'Model training' section. We then calculated the the prediction accuracy on the full data set. We repeated this procedure five times for each model and fit a Bayesian linear model to a randomly selected subset of the evaluation data to compare the results statistically (see Appendix 7).

These results demonstrate that our re-implementation of and modification to the DeepLabCut model (*Mathis et al., 2018*) have little effect on prediction accuracy (*Appendix 8—figure 1*). We also provide qualitative comparisons of these results in *Appendix 8—figure 1—figure supplement 1* and *Appendix 8—figure 1—video 1*. For these qualitative comparisons, we also added an additional rotational augmentation (drawing from a uniform distribution in the range [−180°, +180°]) when training our implementation of the DeepLabCut model (*Mathis et al., 2018*) as we noticed this improved generalization to the video for situations where the mouse rotated its body axis. To the best of our knowledge, rotational augmentations are not currently available when using the software from *Mathis et al. (2018)*, and *Nath et al. (2019)*, which demonstrates the flexibility of the data augmentation pipeline (*Jung, 2018*) for DeepPoseKit. The inference speed for the odor-trail mouse dataset using our implementation of the DeepLabCut model (*Mathis et al., 2018*) is ~49 Hz with a batch size of 64 (offline speeds) and ~35 Hz with a batch size of 1 (real-time speeds) at full resolution 640×480, which matches well with results from *Mathis and Warren (2018)* of ~47 Hz and ~32 Hz respectively. This suggests our modifications did not affect the speed of the model and that our speed comparisons are also reasonable. Because the training routine could be changed for any underlying model—including the new models we present in this paper—this factor is not relevant when making comparisons as long as training is identical for all models being compared, which we ensure when performing our comparisons.

**Appendix 9**

## Depthwise-separable convolutions for memory-limited applications

In an effort to maximize model efficiency, we also experimented with replacing $3 \times 3$ convolutions in our model implementations with $3 \times 3$ depthwise-separable convolutions — first introduced by *Chollet (2017)* and now commonly used in fast, efficient 'mobile' CNNs (e.g. *Sandler et al., 2018*). In theory, this modification should both reduce the memory footprint of the model and increase inference speed. However we found that, while this does drastically decrease the memory footprint of our already memory-efficient models, it slightly decreases accuracy and does not improve inference speed, so we opt for a full $3 \times 3$ convolution instead. We suspect that this discrepancy between theory and application is due to inefficient implementations of depthwise-separable convolutions in many popular deep learning frameworks, which will hopefully improve in the near future. At the moment we include this option as a hyperparameter for our Stacked DenseNet model, but we recommend using depthwise-separable convolutions only for applications that require a small memory footprint such as training on a lower-end GPU with limited memory or running inference on a mobile device.

