## [Decision Letter]

Thank you for submitting your article "Fast and robust animal pose estimation" for consideration by *eLife*. Your article has been reviewed by three peer reviewers, including Josh W Shaevitz as a guest Reviewing Editor, and the evaluation has been overseen by Ian Baldwin as the Senior Editor. The following individual involved in review of your submission has also agreed to reveal their identity: Greg Stephens.

The reviewers have discussed the reviews with one another and the Reviewing Editor has drafted this decision to help you prepare a revised submission.

Summary:

This is a very well written resource article that covers the role of deep-learning in animal pose estimation, develops a new method with several improvements in accuracy and speed, and compares this method to existing methods in the literature. This is a timely paper and the improvements are likely to have a significant impact on users in this field. While this field is changing extremely rapidly, the reviewers believe that this paper will both move the technology further and also provide a readable review to the field for newcomers. However, the reviewers identified several issues that need to be addressed before publication, including text not well explained or a bit exaggerated, the effect of relatively small datasets, training routine differences that might affect the comparisons made, and a lack of explanation of the data acquisition.

Essential revisions:

1) Issues with the code:

a) In the script deepposekit/augment/__init__.py, the line 'from. import augmenters' needed to be substituted by 'from imgaug import augmenters'.

b) The module imgaug had to be installed.

c) We also found that comments in code were good at the beginning but less detailed later.

d) The example notebook "step5_predict.ipynb" could use some more instruction. In particular, what is missing is a section of code to analyze the full video.avi file, instead of just one batch of 5000 frames, which might be confusing for a beginner.

e) One suggestion for "step_4_train_model.ipynb". In the section "Define callbacks to enhance model training", the kwarg for the Logger object should be renamed "validation_batch_size" instead of "batch_size", since it is indeed using validation frames. If one labels a small number of annotated examples, then it is possible to get an error here, as the logger will try and use more validation frames than are actually available. The renaming of this variable might help any confusion.

2) Subsection “Animal pose estimation using deep learning”, fourth paragraph: I would recommend writing more text on the distinction between single and multi-animal pose estimation and tracking in the main text. This is a very important issue and I worry that the casual/uninitiated reader might be confused and not look at Appendix 4. For some systems, tracking is very difficult and it should be clear to readers that this method will be difficult to use out-of-the-box for those systems.

3) The Abstract and title do not specifically mention the key novelties of the manuscript and should be rewritten.

4) 'Further details of how these image datasets were acquired, preprocessed, and tracked before applying our pose estimation methods will be described elsewhere.' I think they need to be given here.

5) How do the presented methods differ depending on the amount of labelled data? In the subsection “Experiments and model comparisons”, the authors postulate that differences in methods depending on training routines are minimal. As you are proposing an improvement over these methods, you need to prove this. You should also add a discussion of how many frames one should annotate before starting. While this is an incremental process (using the network to initialize further annotations), how many frames should one label at first? Also, as a related point, how does the accuracy of the network depend on the number of annotations?

6) It is apparent that machine vision methods to track animal behavior on the scale of posture will continue to advance at a remarkable speed. The authors could add substantial and long-lasting value to their work by discussing some of the more general aspects of behavioral measurement. Some possibilities:

a) It was only a few years ago that most behavioral measurements focused on image centroids and it would be useful to expand on the usefulness of representing behavior through posture vs. centroid.

b) What behavioral conditions remain challenging for the current generation of pose estimation algorithms (including DeepPoseKit)? For example, it would seem that a densely-packed fish school or bee hive might require novel approaches for both individual identity, the 3D nature of the school and resulting occlusions. This is an important consideration for the comparison of techniques. For example LEAP was designed very directly for high-contrast, controlled laboratory environments and it is perhaps not surprising that LEAP fares worse under less ideal conditions.

c) Relatedly, when would we consider the "pose tracking" problem solved? For example, the number of body points is user- not organism-defined, when do we know that we have enough?

d) The DeepPoseKit algorithm leverages multiple spatial scales in the input image data and it would be useful to expand the discussion about why this is beneficial. For example, for the fly data, what explicitly are the multiple scales that one might want to learn from the images? Can you further discuss how exactly does multi-scale inference achieve the fast speed of LEAP without sacrificing accuracy

e) With deep learning algorithms especially, there is a danger of rare but very wrong label assignments. Since DeepPoseKit is designed for general use, including among those not experienced in such networks, it would be quite useful to emphasize post-processing analysis that can help minimize the effect of these errors.

7) The manuscript would benefit from a discussion of how long it takes to train the networks and especially interesting would be a benchmarking of the three algorithms: DeepPoseKit, LEAP and DeepLabCut.

---

## [Author Response]

Essential revisions:1) Issues with the code:a) In the script deepposekit/augment/__init__.py, the line 'from. import augmenters' needed to be substituted by 'from imgaug import augmenters'.

We thank the reviewers for pointing out this oversight. The __init__.py file has already been updated since the initial release to correct this bug. This was not actually related to imgaug (although the described substitution does solve the import error) but was related to legacy code from when we originally developed our own data augmentation pipeline before switching to the imgaug package.

b) The module imgaug had to be installed.

It was brought to our attention by other users that imgaug needs to be manually installed when using Anaconda on Windows (and potentially other operating systems). We have updated the README with additional details that imgaug should be manually installed when using Anaconda (https://github.com/jgraving/deepposekit/blob/master/README.md#installation). We are working to address these issues with Anaconda as best as possible. Otherwise the imgaug module should be installed automatically as a dependency when installing DeepPoseKit with pip using the README instructions, which we have tested with many other systems. This has been included in the setup.py script since the initial public release of the code.

c) We also found that comments in code were good at the beginning but less detailed later.

This is an excellent point. We provided only minimal documentation in order to be able to send the code to the reviewers as quickly as possible and avoid further delays with the review process. We have further updated our example notebooks with more extensive comments as suggested. We have also added more doc strings to classes and functions to improve the general documentation. Adding additional documentation to the code will take time and effort, but we are working to address this as best as possible for future updates.

d) The example notebook "step5_predict.ipynb" could use some more instruction. In particular, what is missing is a section of code to analyze the full video.avi file, instead of just one batch of 5000 frames, which might be confusing for a beginner.

We have updated this notebook with an example for processing an entire video and saving the data to disk with more extensive comments to explain the details of the code.

e) One suggestion for "step_4_train_model.ipynb". In the section "Define callbacks to enhance model training", the kwarg for the Logger object should be renamed "validation_batch_size" instead of "batch_size", since it is indeed using validation frames. If one labels a small number of annotated examples, then it is possible to get an error here, as the logger will try and use more validation frames than are actually available. The renaming of this variable might help any confusion.

We thank the reviewers for this excellent suggestion. We have updated the code as specified.

2) Subsection “Animal pose estimation using deep learning”, fourth paragraph: I would recommend writing more text on the distinction between single and multi-animal pose estimation and tracking in the main text. This is a very important issue and I worry that the casual/uninitiated reader might be confused and not look at Appendix 4. For some systems, tracking is very difficult and it should be clear to readers that this method will be difficult to use out-of-the-box for those systems.

We have updated the main text to more clearly and thoroughly make the distinction between individual and multiple pose estimation (subsection “Individual vs. multiple pose estimation”). We have also added a discussion of the advantages and disadvantages of using tracking and individual annotations vs. no tracking and multiple (exhaustive) annotations of full-sized images. This should help to make clear to the reader that our method may be difficult to use for some systems where tracking is difficult or not possible.

3) The Abstract and title do not specifically mention the key novelties of the manuscript and should be rewritten.

We have updated the title and modified the Abstract to explicitly mention the key novelties presented in the manuscript.

4) 'Further details of how these image datasets were acquired, preprocessed, and tracked before applying our pose estimation methods will be described elsewhere.' I think they need to be given here.

We have updated the subsection “Datasets” to provide a more detailed description of our image acquisition, tracking, and preprocessing procedures. The tracking algorithms used for our datasets are unpublished and would take significant space to describe in full detail. Adding this description is outside the scope of this paper and would take away from the main focus of our pose estimation methods. These tracking algorithms will also be the subject of further publications and we do not wish to reduce the novelty of these publications. The details of different localization and tracking methods are not especially relevant for comparing pose estimation algorithms other than the fact that individuals are successfully localized and tracked before cropping and annotating. Any of the many already-available tracking algorithms cited in the paper could be used for this preprocessing step, and of course, each has its own set of advantages and disadvantages that are not relevant to this paper.

5) How do the presented methods differ depending on the amount of labelled data?Also, as a related point, how does the accuracy of the network depend on the number of annotations?

We assume this is the same question, otherwise please let us know if these are distinct questions that should be addressed separately. We have performed additional experiments and updated the text to address these comments. Appendix 1—figure 3 shows that our methods need little training data to generalize well to new data. Subsection “Stacked DenseNet trains quickly and requires few training examples”, first paragraph in the main text provide further details of these results.

In the subsection “Experiments and model comparisons”, the authors postulate that differences in methods depending on training routines are minimal. As you are proposing an improvement over these methods, you need to prove this.

To address this we adapted one of the example datasets from Mathis et al., 2018 to work with DeepPoseKit and then directly compared the two training routines (from the original paper and our modified implementation). We find that there is effectively no difference in prediction accuracy between our implementation and the original implementation from Mathis et al., 2018 when data are augmented in the same way during training. We have added Appendix 8—figure 1 and discussion in Materials and methods second paragraph and Appendix 8 subsection “Our implementation of the DeepLabCut model” to address this point. Additionally we provide a video of the posture tracking output for a novel video from this dataset (Appendix 8—figure 1—video 1) and plots of the time series output (Appendix 8—figure 1—figure supplement 1) for qualitative comparison. Together these results demonstrate that our implementation of the DeepLabCut model actually generalizes slightly better to novel data.

You should also add a discussion of how many frames one should annotate before starting. While this is an incremental process (using the network to initialize further annotations), how many frames should one label at first?

We have added discussion of this in the last paragraph of the subsection “Stacked DenseNet trains quickly and requires few training examples” in relation to how much training data is required for the model to generalize well.

6) It is apparent that machine vision methods to track animal behavior on the scale of posture will continue to advance at a remarkable speed. The authors could add substantial and long-lasting value to their work by discussing some of the more general aspects of behavioral measurement. Some possibilities:a) It was only a few years ago that most behavioral measurements focused on image centroids and it would be useful to expand on the usefulness of representing behavior through posture vs. centroid.

We have expanded on the discussion of general aspects of measuring behavior including this point in the subsection “Measuring animal movement with computer vision”.

b) What behavioral conditions remain challenging for the current generation of pose estimation algorithms (including DeepPoseKit)? For example, it would seem that a densely-packed fish school or bee hive might require novel approaches for both individual identity, the 3D nature of the school and resulting occlusions. This is an important consideration for the comparison of techniques. For example LEAP was designed very directly for high-contrast, controlled laboratory environments and it is perhaps not surprising that LEAP fares worse under less ideal conditions.

We have added discussion of this in the sixth and eighth paragraphs of the Discussion.

c) Relatedly, when would we consider the "pose tracking" problem solved? For example, the number of body points is user- not organism-defined, when do we know that we have enough?

We have added discussion of this in the eighth paragraph of the Discussion.

d) The DeepPoseKit algorithm leverages multiple spatial scales in the input image data and it would be useful to expand the discussion about why this is beneficial. For example, for the fly data, what explicitly are the multiple scales that one might want to learn from the images? Can you further discuss how exactly does multi-scale inference achieve the fast speed of LEAP without sacrificing accuracy.

We have added additional discussion of this point in the last paragraph of the subsection “Pose estimation models and the speed-accuracy trade-off” and further discussion can be found in Appendix 4.

e) With deep learning algorithms especially, there is a danger of rare but very wrong label assignments. Since DeepPoseKit is designed for general use, including among those not experienced in such networks, it would be quite useful to emphasize post-processing analysis that can help minimize the effect of these errors.

We have added discussion of this in in the last paragraph of the subsection “Animal pose estimation using deep learning” and in the first paragraph of the subsection “Learning multi-scale geometry between keypoints improves accuracy and reduces extreme errors” and updated Figure 1 to better emphasize this point.

7) The manuscript would benefit from a discussion of how long it takes to train the networks and especially interesting would be a benchmarking of the three algorithms: DeepPoseKit, LEAP and DeepLabCut.

We have performed additional experiments and added discussion of this point. See subsection “Stacked DenseNet trains quickly and requires few training examples” and Appendix 1—figure 3.